# Functional and biochemical characterization of the *Toxoplasma gondii* succinate dehydrogenase complex

**Mariana F. Silva, Kiera Douglas, Sofia Sandalli, Andrew E. Maclean** *, **Lilach Sheiner** *

Wellcome Centre for Integrative Parasitology, University of Glasgow, Glasgow, United Kingdom

* andrew.maclean@glasgow.ac.uk (AEM); lilach.sheiner@glasgow.ac.uk (LS)

**Data Availability Statement:** All data are in the manuscript and/or supporting information files.

## Abstract

The mitochondrial electron transport chain (mETC) is a series of membrane embedded enzymatic complexes critical for energy conversion and mitochondrial metabolism. In commonly studied eukaryotes, including humans and animals, complex II, also known as succinate dehydrogenase (SDH), is an essential four-subunit enzyme that acts as an entry point to the mETC, by harvesting electrons from the TCA cycle. Apicomplexa are pathogenic parasites with significant impact on human and animal health. The phylum includes *Toxoplasma gondii* which can cause fatal infections in immunocompromised people. Most apicomplexans, including *Toxoplasma*, rely on their mETC for survival, yet SDH remains largely understudied. Previous studies pointed to a divergent apicomplexan SDH with nine subunits proposed for the *Toxoplasma* complex, compared to four in humans. While two of the nine are homologs of the well-studied SDHA and B, the other seven have no homologs in SDHs of other systems. Moreover, SDHC and D, that anchor SDH to the membrane and participate in substrate bindings, have no homologs in Apicomplexa. Here, we validated five of the seven proposed subunits as bona fide SDH components and demonstrated their importance for SDH assembly and activity. We further find that all five subunits are important for parasite growth, and that disruption of SDH impairs mitochondrial respiration and results in spontaneous initiation of differentiation into bradyzoites. Finally, we provide evidence that the five subunits are membrane bound, consistent with their potential role in membrane anchoring, and we demonstrate that a DY motif in one of them, SDH10, is essential for complex formation and function. Our study confirms the divergent composition of *Toxoplasma* SDH compared to human, and starts exploring the role of the lineage-specific subunits in SDH function, paving the way for future mechanistic studies.

## Author summary

Apicomplexans, such as *Toxoplasma gondii*, are parasites of humans and animals that cause diseases of global importance, such as toxoplasmosis which can be fatal to the immunocompromised. The mitochondrial electron transport chain consists of a series of enzymatic complexes that are needed by the parasite for energy metabolism and for

**Funding:** Our work was funded by research grants from the Medical Research Council MR/W002221/1 to LS and AM, the Wellcome Trust UNS85804 to LS and the Wellcome Trust 221681/Z/20/Z to AM. The funders had no role in study design, data collection and analysis, decision to publish, or preparation of the manuscript.

**Competing interests:** The authors have declared that no competing interests exist.

producing important metabolites. The compositions of the chain and its enzymes are different between parasites and their human host. Here, we study the succinate dehydrogenase complex (complex II) of the chain and confirm that the *Toxoplasma* enzyme has different subunits compared to human and that they are critical for its function. We further start unravelling the role of the parasite subunits within the enzyme's mechanism of action. The mitochondrial electron transport chain is an effective drug target for *Toxoplasma* and other related pathogens such as the malaria causing parasite. We show here that the complex II is important for parasite growth, and validate its differences from the human version, thus enhancing our understanding of this critical parasite pathway.

## Introduction

The mitochondrial electron transport chain (mETC) is composed of a series of protein complexes embedded in the mitochondrial inner membrane that transfer electrons harvested by different mitochondrial enzymes, through the complexes, to oxygen. The mETC activity is critical for cell survival as it enables or regulates essential metabolic pathways such as the tricarboxylic acid (TCA) cycle, pyrimidine biosynthesis and mitochondrial ATP synthesis. Due to its importance, the mETC has been extensively studied in common model eukaryotes, primarily in yeast and mammals, and the composition and function of its complexes have been characterized in detail in those systems [1–3]. More recently, as a broader repertoire of organisms are being studied, divergence in the composition of the complexes and in their structural and regulatory features is gradually exposed [4–8], highlighting a need to understand how the mETC works in a wide range of eukaryotic models. Insights from such studies will deepen our understanding of the evolution of divergent organisms including pathogenic parasites.

Apicomplexan parasites are single cell eukaryotes that are divergent from the ophistokont clade of eukaryotes to which yeast and mammals belong. As such, an understanding of apicomplexan mitochondrial biology makes an important contribution to our knowledge of eukaryotic life. Additionally, these obligate intracellular parasites are responsible for numerous human and animal diseases with substantial global impact. Particularly, *Plasmodium falciparum* causes hundreds of thousands of deaths each year from human malaria, primarily in lower income countries, and *Toxoplasma gondii*, causes life threatening infections in immunocompromised patients around the globe. Despite their parasitic lifestyle, the apicomplexans that possess an mETC depend on the transport of electrons throughout their complicated life cycles, and the activity of some of their mETC complexes is required for survival, dissemination, and transmission [9–18]. Due to these essential roles, the mETC is a target for anti-parasitic drugs, with a prime example provided by the antimalarial atovaquone that targets complex III (Cytochrome $bc_1$ complex) [19–21]. Despite this potential, the details of how the apicomplexan mETC complexes work remain mostly unknown.

Recent studies have explored the compositions of the apicomplexan mETC complexes via different biochemical and proteomics methods [10,13,14,22]. These studies suggested that each of the mETC complexes contain subunits for which homologs are not clearly identifiable, and likely absent, in yeast and mammals. The mETC complex which showed the most differences in size, subunit number and subunit identity is complex II. For example, in *T. gondii*, our previous complexome profiling data predicted complex II to contain 9 subunits and migrate at 500 kDa [13]. These findings are supported by studies in other apicomplexans [22,23] together pointing to a complex that is much larger than the ~130 kDa and 4-subunits

seen in mammals and yeast and even in the bacterial ancestor of mitochondria. However, these finding are yet to be validated and the new subunits are yet to be functionally studied.

Complex II, also known as succinate dehydrogenase (SDH), catalyses the oxidation of succinate to fumarate, thereby harvestings electrons from the TCA cycle and then feeding them into the mETC. As such, SDH is a major link between the mETC and the TCA cycle, it is thus a highly important player in the coordination between mitochondrial metabolic pathways [24]. Moreover, alongside a series of dehydrogenases, SDH is one of the first points of entry for electrons into the chain. In the reaction catalysed by SDH, the electrons harnessed from succinate are transferred, via a FAD cofactor and three iron-sulfur (2Fe-2S) clusters, to reduce ubiquinone to ubiquinol, which then shuttles electrons onward to Complex III. The ubiquinone binding sites therefore play a critical role in the overall catalytic reaction of SDH.

In ophistokonts and in bacteria SDH is composed of four subunits (SDHA-D, or SDH1-4) which together form a complex that could be broadly divided into two main domains: a matrix localized enzymatic core and a membrane anchoring domain. In mammals, SDHA and SDHB make up the matrix facing domain, which contains the succinate binding site, the FAD cofactor (both within SDHA), and the three 2Fe-2S clusters (within SDHB). SDHC and SDHD form the membrane anchoring domain and contain a heme b prosthetic group [25]. The two ubiquinone binding sites are formed through interactions that are contributed to by amino acids from multiple subunits. One, high affinity, site ($Q_P$-proximal) lies on the matrix side of the inner membrane and is formed by residues from SDHB, C and D, and another, low affinity, site ($Q_d$-distal) is closer to the inter membrane space [25]. In apicomplexans, homologs of SDHA and SDHB are readily identifiable via sequence-based searches [26,27]. However, no clear homologs of SDHC and SDHD could be found using homology searches. It is thus tempting to propose that the putative new subunits found in the complexome studies may replace the roles of SDHC and SDHD. Here we aimed to validate the new putative subunits of *T. gondii* SDH, to examine their importance for parasite growth and complex assembly and function, and to elucidate their role within the complex.

## Results

### Localization and native migration validate five new *Toxoplasma* SDH subunits

Our previous complexome profiling study [13] identified seven putative novel SDH subunits based on their co-migration with a homolog of the known subunit SDHB (**Table 1**: summarizes the putative subunits along with their proposed names). Homologs of five of these seven proteins were also identified in a similar complexome study in *Plasmodium falciparum* [22]. We sought to validate these predictions. We reasoned that all new SDH subunits would be found in the mitochondrion. Data from a previous Hyper Localisation of Organelle Proteins by Isotope Tagging (HyperLOPIT) study in *Toxoplasma* [28], as well as a mitochondrial matrix proteome [10], supports mitochondrial localization of five of the seven new subunits [13]. To confirm the mitochondrial localization of these five and to assess the localization of the two additional candidates, we transiently expressed an ectopic Ty epitope tagged copy of each of the seven proteins in the parasites. We then performed immunofluorescence assays where we co-stained for Ty and the mitochondrial marker TgMys [29]. All subunits co-localized with the mitochondrial marker, indicating a mitochondrial localization (**Fig 1A**). We performed super-resolution microscopy to examine the localization of putative SDH15 and the outer mitochondrial membrane proteins TOM40 [30]. Overlay of the signals provides support for mitochondrial localization. (**S1A Fig**). Since SDH is embedded in the inner mitochondrial membrane we expected its subunits to localize to this compartment. We performed ultra-

**Table 1. Details of putative SDH subunits.** Phenotype scores were taken from [31]. Names for novel subunits were given the prefix SDH followed by the approximate molecular weight, except for those previously named*[63]. Transmembrane domains (TMD) were predicted using PRED-TMR [64], CCTOP [65] and TMbed [66] and the number of predicted TMDs displayed.

| Gene ID | Name | Phenotype score | Size (kDa) | TMD | | |
|---|---|---|---|---|---|---|
| | | | | PRED-TMR | CCTOP | TMbed |
| TGGT1_215590 | SDHA | -3.96 | 72.7 | 1 | 0 | 0 |
| TGGT1_215280 | SDHB | -2 | 38.6 | 0 | 0 | 0 |
| TGGT1_227920 | SDH10 | -2.42 | 9.5 | 1 | 1 | 1 |
| TGGT1_226500 | SDH11 | -0.89 | 11.1 | 1 | 1 | 1 |
| TGGT1_252630 | SDH15 | -1.45 | 15.1 | 0 | 0 | 1 |
| TGGT1_315930 | SDH18 | -3.84 | 18.1 | 1 | 2 | 3 |
| TGGT1_206480 | SDH23 | -1.8 | 22.5 | 1 | 2 | 1 |
| TGGT1_306650 | SDH31 | -1.8 | 31.4 | 0 | 0 | 0 |
| TGGT1_223485 | MPODD* | -1.49 | 10.3 | 0 | 0 | 0 |

expansion microscopy to examine the localization of putative SDH15 and of SDHB compared to the outer mitochondrial membrane proteins TOM40 (**Fig 1B**). Overlay of the signals suggests that SDH15 and SDHB are found in a different compartment from TOM40 (**S1B Fig**), consistent with localization in the inner and outer mitochondrial membranes respectively.

We further reasoned that any SDH component would migrate at ~500 kDa, as they did in the complexome [13]. For this we constructed stable lines where an ectopic Ty epitope tagged copy of each of the proteins is stably expressed. We generated stable lines with Ty tagging of genes encoding for five of the putative subunits: SDH10,11,15,23 or 31, and immunoblot analysis confirmed that the ectopic Ty-tagged proteins are detected at their predicted sizes (**S2A Fig**). For each line we solubilized parasite cells in the non-ionic detergent Dodecyl-beta-D-maltoside (βDDM) and performed blue native PAGE followed by immunoblotting with anti-Ty antibody. While in the case of putative SDH11 the signal on native PAGE was insufficient to resolve its migration via immunoblotting, the four other lines showed a single band at ~500 kDa (**Figs 1C and S1D**), consistent with the migration of the SDHB-HA control and as previously shown for SDHB-HA [13]. These observations demonstrate that at least the four subunits tested migrate as part of a large molecular weight complex, consistent with them being SDH subunits.

To provide direct evidence for an interaction between one of the new subunits and the conserved subunit SDHB, we performed reciprocal co-IP. We constructed a line where putative SDH15 is expressed as an ectopic Ty epitope tagged copy, in a background where SDHB is endogenously HA tagged [13]. We then performed an IP with HA agarose beads. In the bound fraction both SDHB-HA and the putative SDH15-Ty were recovered (**Fig 1D**). Neither was found in the unbound fraction and the unrelated mitochondrial protein TOM40 was only found in the unbound fraction, providing controls for the co-IP specificity. Performing the reciprocal IP with Ty beads returned the same result, with both SDHB-HA and putative SDH15-Ty recovered in the bound fraction. These results demonstrate that SDHB and putative SDH15 interact, or have common interaction partners, suggesting they are both part of the same complex. Finally, to provide further evidence of interaction we performed IPs of SDHB-HA and complex III's QCR2-HA [13] and sent the elutions for mass spectrometry analysis. We identified eight of the nine putative subunits as being enriched in the SDHB-HA IP compared to the QCR2-HA IP across four independent experiments (Figs 1E, **S3A, S3B and S2 Table**, PXD047027). Conversely, we detected six complex III subunits that were enriched in the QCR2-HA IP compared to the SDHB-HA-IP. These data further support the new

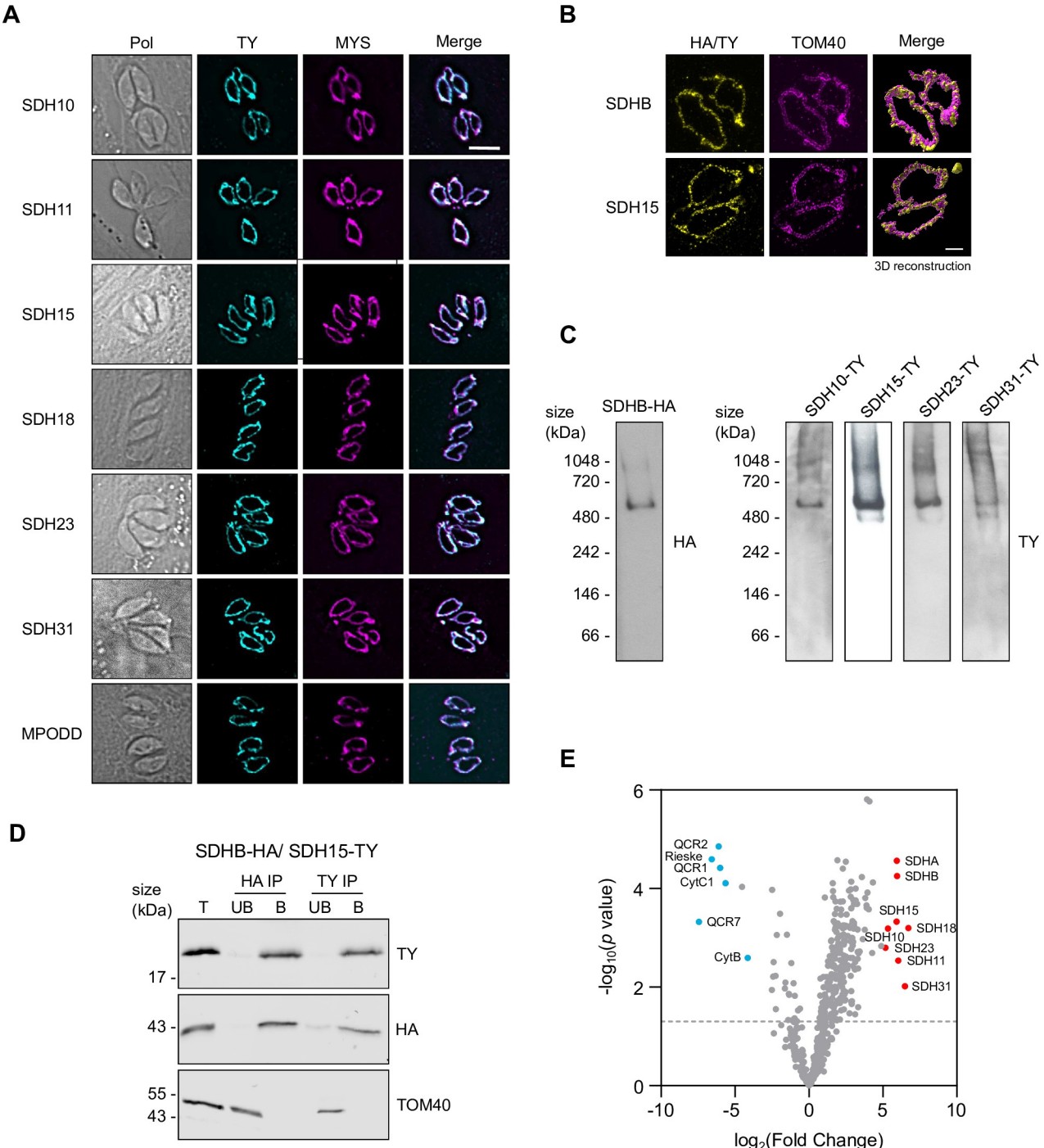

**Fig 1. Putative *Toxoplasma* SDH subunits localize to the mitochondrion, are part of a ~500 kDa complex and SDH15 interacts with SDHB.** (A) Immunofluorescence assay analysis of putative SDH subunits transiently tagged with a Ty epitope tag (cyan), showing co-localization with the mitochondrial marker protein MYS (TGME49_215430, [29]) (magenta). Scale bar is 5 µM. (B) Ultra-expansion microscopy images shown as Z-stack (Left and middle panel) and 3D reconstruction (right panel) of SDHB-HA and SDH15-Ty, with the mitochondrial marker protein TOM40. Scale bar is 3 µM. (C) Total lysate from Ty epitope tagged putative SDH subunits and SDHB-HA separated by BN-PAGE, blotted and immunolabelled with either anti-Ty or anti-HA antibodies. All samples were separated on the same gel, different exposures shown for different samples for clarity. The full unprocessed gel in **S1D Fig**. (D) Immunoblot analysis of whole cell lysate extracted from rSDH15/ SDHB-HA + SDH15-Ty and immunoprecipitated with anti-HA or anti-Ty antibody coupled beads, to produce total lysate (T), unbound (UB) and bound (B) fractions. Samples were separated by SDS-PAGE, blotted, and detected using anti-HA and anti-Ty antibodies to label immunoprecipitated proteins, and anti-TOM40 as an unrelated mitochondrial protein control. (E) Volcano plot of proteomic data from SDHB-HA and QCR2-HA immunoprecipitations, showing the -$\log_{10}$ $p$-values and the $\log_2$ fold changes of proteins detected by mass spectrometry across four independent experiments. Complex III subunits are labelled in blue and complex II subunits are labelled in the red. Dotted grey line denotes $p < 0.05$.

subunits as belonging to the same complex as the conserved complex II subunits SDHB and SDHA.

## The putative new subunits are required for SDH stability and function

The evidence of mitochondrial localization and migration at a size compatible with *Toxoplasma* SDH provided support for the candidate proteins being part of this complex. We therefore explored the contribution of each protein to SDH assembly and function. For this, we turned to generating mutants of each of the five putative components, SDH10,11,15,23 or 31. Previous work demonstrated that components of the mETC complexes of *Toxoplasma* are typically important for parasite fitness, and mutants of subunits of complex III and IV are unable to grow in standard culture conditions [10,13,14,16], suggesting that SDH subunits may also be important for fitness. Consistent with this expectation, the *Toxoplasma* genome wide CRISPR screen for growth in culture predicts that all five genes likely contribute to parasite fitness [31]. We therefore opted to generate conditional mutants for each of them. Using a parental *T. gondii* line whereby SDHB is endogenously HA tagged [13] as background, conditional knock-down lines for each of the five genes were generated via our promoter replacement strategy (**S2B–S2C Fig**) whereby the addition of anhydro-tetracycline (ATc) results in down-regulation of the gene of interest [32]. We refer to those lines as "r(subunit name)/SDHB-HA" (r for regulatable). qRT-PCR confirmed that the expression of each of the five genes is significantly down-regulated upon addition of ATc for two days (**S2D Fig**). Each line was also complemented with a stably integrated Ty epitope tagged version of the gene under the control of a constitutive promoter, which is not regulated by ATc, to serve as control. We refer to those lines as "r(subunit name)/SDHB-HA–(subunit name)-Ty".

Using those mutants, we examined whether depletion of any of the new subunits affected the stability of SDH. After culturing rSDH15/SDHB-HA parasites in the absence or presence of ATc for one, two, or three days we performed blue native PAGE followed by immunoblotting with anti-HA antibodies to track the migration of SDHB-HA, which acts as an indicator for the fully formed SDH. Loss of the high molecular weight band at ~500 kDa is seen on day two, suggesting there is no, or very low levels, of mature complex formed at this time point (**Fig 2A**). This is despite the fact that SDHB-HA could still be detected in a denaturing immunoblot, albeit at reduced levels (**Fig 2A**). As a control, treatment of the parental line SDHB-HA with ATc for three days had no effect on SDHB native migration (**Fig 2B**). Likewise, when the same experiment was performed with the complemented line, rSDH15/SDHB-HA–SDH15-Ty, the SDHB-HA high molecular weight band was still visible after three days of ATc treatment (**Fig 2C**). These data suggest that the putative SDH15 is required for SDH stability. The reduction in SDHB-HA levels upon knockdown of these other subunits raises a possibility that SDHB stability might depend on complex integrity. To control for a general mitochondrial defect in this mutant, blue native PAGE and immunoblotting of the protein import translocon component, TOM40, as well as clear native PAGE followed by an in-gel complex IV activity assay were performed. We found that both the TOM complex and Complex IV are unaffected by the depletion of the putative SDH15, suggesting that its depletion results in a SDH specific defect (**Fig 2D–2E**). In support of a specific SDH defect mitochondrial morphology is also unchanged in this mutant, unlike what is found in another mitochondrial mutant with a known effect on mitochondrial morphology (VDAC depletion [33]) used as a control (**S4 Fig**). Native migration experiments were performed for the other four subunits. In all cases depletion of the subunit at day three of ATc treatment resulted in disappearance of the 500 kDa SDHB-HA band, while a band was still visible in denaturing immunoblot, and this complex stability defect was rescued in the complemented lines (**Fig 2F**). As control again we

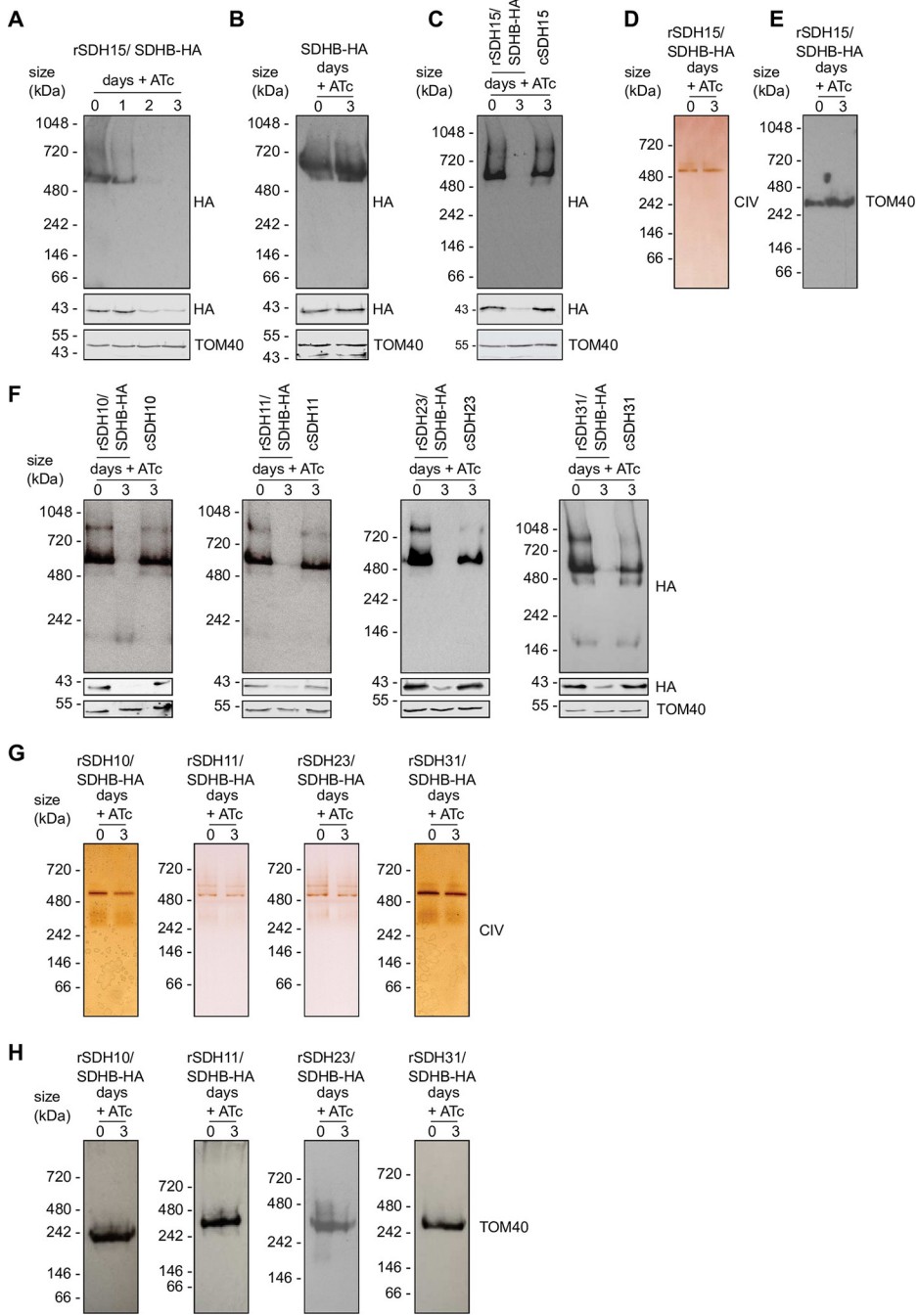

**Fig 2. Putative *Toxoplasma* SDH subunits are important for its stability.** (A) Total lysate from rSDH15/ SDHB-HA parasites grown in the absence (0) or presence of ATc for one-three days were separated by BN-PAGE and immunoblot analysis performed with anti-HA antibodies. Samples were also separated by SDS-PAGE and immunolabelled with anti-HA and anti-TOM40 antibodies. (B) Total lysate from SDHB-HA grown in the absence or presence of ATc for three days were separated by BN-PAGE and SDS-PAGE, and immunolabelled as in *(A)*. (C) Total lysate from rSDH15/ SDHB-HA parasites grown in the absence (0) or presence of ATc for three days, and rSDH15/ SDHB-HA -SDH15-Ty (cSDH15) grown in the presence of ATc for three days separated by BN-PAGE and SDS-PAGE and immunolabelled, as in *(A)*. (D) Total lysate from rSDH15/ SDHB-HA parasites grown in the absence (0) or presence of ATc for three days were separated by clear-native PAGE and stained for complex IV activity. (E) Total lysate from rSDH15/ SDHB-HA parasites grown in the absence (0) or presence of ATc for three days, separated by BN-PAGE and immunoblotted with anti-TOM40 antibodies. (F-H) rSDH10,11,23 or 31/ SDHB-HA and complemented lines grown in the absence (0) or presence of ATc for three days and separated by native PAGE as described in *A*, *C* and *D*.

found that both the TOM complex and Complex IV are unaffected by the depletion of any of the four subunits (**Fig 2G and 2H**), suggesting that their depletion results in a SDH specific defect.

Next, we performed a spectrophotometric enzymatic assay to determine the activity of the SDH upon depletion of each new subunit. Lysates from ATc treated and untreated mutants as well as treated complemented lines were incubated with the substrate succinate and the ubiquinone analogue decylubiquinone (DUB). Succinate oxidation by SDH results in reduction of DUB, which, in turn, reduces the redox dye 2,6-dichlorophenolindophenol (DCPIP), which then changes colour providing a proxy for SDH activity. In each of the five mutants, treatment with ATc for three days resulted in decreased activity, and this defect was rescued in the complemented lines (**Fig 3A** and **S3 Table**). As a control for the specificity of the assay for SDH activity, rather than a general defect in parasite metabolism, we performed activity assays for the dehydrogenase MQO, which uses malate as a substrate rather than succinate. Lysate was supplied with malate instead of succinate, and we observed that depletion of each of the subunits did not affect the enzymatic activity as expected (**Fig 3B** and **S3 Table**). These observations demonstrate that the new proteins are required for SDH enzymatic activity.

Previous studies into the *Toxoplasma* mETC found that mutants in mETC complexes resulted in a decrease in mitochondrial oxygen consumption rate (mOCR), which is used as a proxy measurement for respiratory activity [10,14]. As one of the entry points for electrons into the mETC, it is expected that SDH disruption would result in a respiration defect. We again focused on the putative SDH15 and examined the effect of its depletion on the mOCR with a seahorse extracellular flux analyzer, using a previously established assay [10,34]. We initially measured the basal mOCR of parasites grown in the presence or absence of ATc for three days. We detected no difference in the basal mOCR in the parental line, but in the depletion mutant we saw a significant decrease and this defect was rescued in the complemented line (**Fig 3C(i)**). A similar pattern was seen when measuring the maximal mOCR (**Fig 3C(ii)**), which is done after adding the protonophore FCCP, which decouples oxygen consumption form ATP synthesis. We simultaneously measured the extracellular acidification rate (ECAR) as a proxy for general parasite metabolism. This was unaffected upon the putative SDH15 depletion (**Fig 3C(iii)**), suggesting the observed decreased mOCR is due to a specific defect in the mETC, and not a general decrease in parasite metabolism or viability. These data suggest that the putative SDH15 is important for mitochondrial oxygen consumption, although the magnitude of the mOCR decrease is less pronounced than for complex III and complex IV mutants [10,14], possibly due to the compensatory action of other dehydrogenases.

Taken together, the localization, migration and necessity for complex stability and function provides a validation that the putative SDH10,11,15,23 or 31 are bona fide SDH subunits.

## SDH is important for parasite fitness and its disruption results in spontaneous initiation of differentiation into bradyzoites in culture

SDH is one of the first points of entry of electrons into the mETC, and an important coordinator between the TCA cycle and respiration and ATP synthesis. It was therefore expected that SDH will be important for parasite fitness, as also predicted by the above mentioned CRISPR screen (**Table 1**). To examine this hypothesis, we performed plaque assays. We found that depletion of SDH10,11, 15, 23 and 31 result in a severe growth defect leading to highly reduced plaque size (**Fig 4A and 4B**). This phenotype was rescued in the complemented lines (**Fig 4C**) testifying for the specificity of the observed defect. Thus, while all the new subunits contribute to fitness, they may be not strictly essential, or the knockdown may be insufficiently stringent to fully ablate SDH activity. To further characterize the importance of SDH for parasite growth

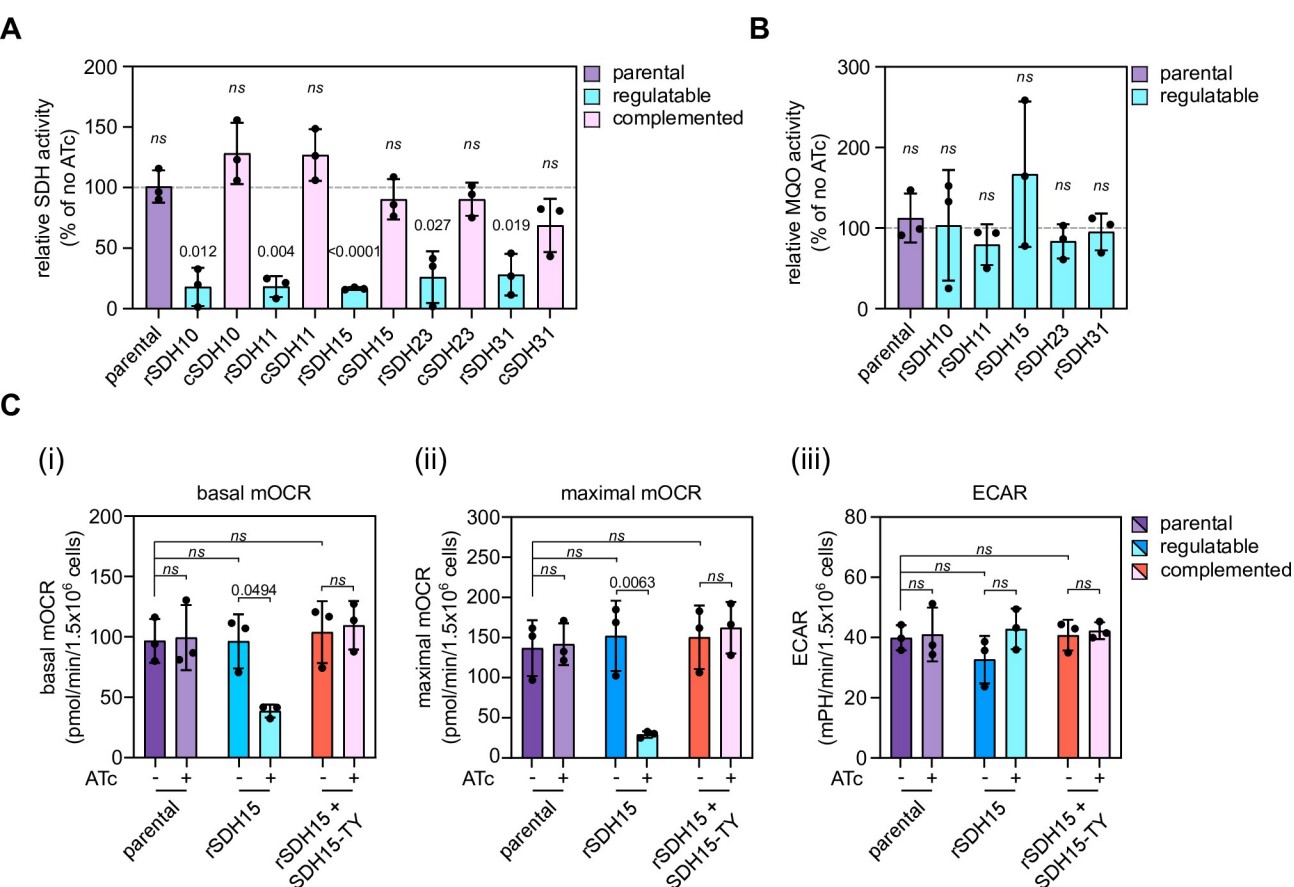

**Fig 3. Depletion of putative SDH subunits leads to decreased SDH activity and oxygen consumption.** (A) SDH activity of parental, rSDH10,11,15,23 or 31/ SDHB-HA and complemented parasites, grown in the presence or absence of ATc for three days. Graphs show mean relative SDH activity -/+ S.D. from three independent biological replicates. No ATc was set to 100% and plus ATc conditions were compared by a one-sample t-test. SDH activity values are in **S3 Table**. (B) MQO activity of parental, rSDH10,11,15,23 or 31/ SDHB-HA parasites, grown in the presence or absence of ATc for three days. Graphs show mean relative MQO activity -/+ S.D. from three independent experiments. No ATc was set to 100% and plus ATc conditions were compared by a one-sample t-test. MQO activity values in **S3 Table**. (D) Extracellular flux analysis, using a seahorse analyser, of (i) basal mitochondrial oxygen consumption rate (OCR); (ii) maximal mitochondrial OCR and (iii) extracellular acidification rate (ECAR) of SDHB-HA, rSDH15/ SDHB-HA and rSDH15/ SDHB-HA—SDH15-Ty parasites grown in the presence or absence of ATc for 3 days. Graphs show mean -/+ S.D. from three independent biological replicates. ANOVA followed by Turkey's multiple pairwise comparisons was performed, and p-value from relevant pairs displayed.

we performed a replication assay with parental and rSDH11/ SDHB-HA parasites. We grew parasites in the presences or absence of ATc for two days and allowed parasite to invade a HFF monolayer, grow for a further 24 hours in the same conditions, before performing an immunofluorescence assay and assessed parasite replication. We quantified the number or parasites per vacuole and found a significant decrease in the number of vacuoles containing eight or more parasites in the rSDH11 line grown in ATc (**Fig 4D**). This phenotype was not observed in the no ATc control where we observed that 30% of the vacuoles analysed consisted of eight tachyzoites or more. These results demonstrate that the new SDH subunits contribute to parasite growth and replication.

A recent study suggests a link between impairment of mitochondrial functions and the initiation of stage conversion in *Toxoplasma* [35]. Mutants with defects in mitochondrial functions such as respiration, translation, and iron-sulfur (Fe-S) cluster biosynthesis, showed an incomplete differentiation phenotype, forming cyst-like structures, and expressing intermediate bradyzoite markers, but never progressing to mature bradyzoites. To examine if a similar

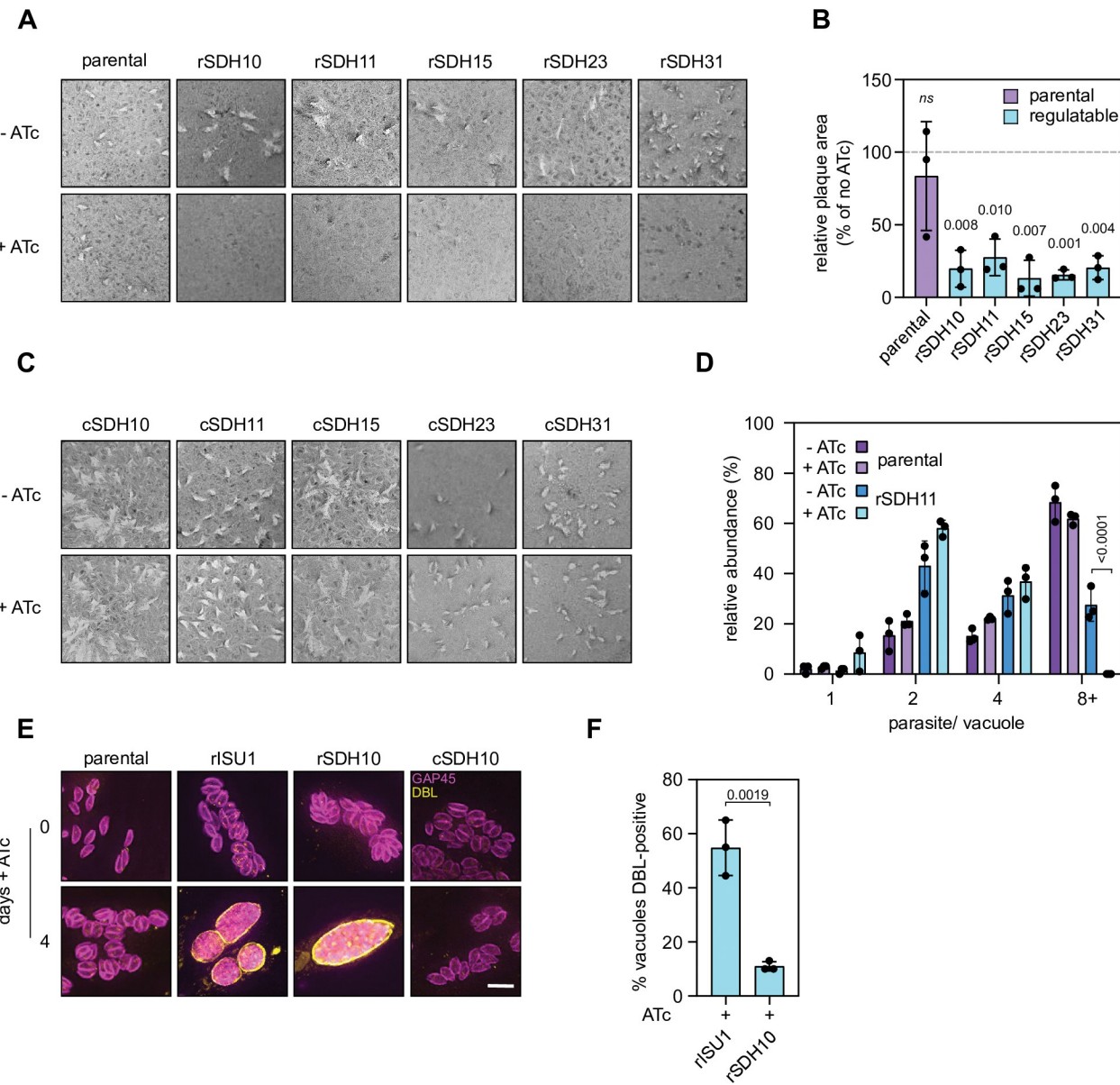

**Fig 4. Novel *Toxoplasma* SDH subunits are important for parasite growth.** (A) Plaque assays of parental and rSDH10,11,15,23 or 31/ SDHB-HA parasites, grown in the presence or absence of ATc for eight days. (B) Quantification of plaque size from *(A)* three independent experiments were performed and 10 plaques measured per replicate. The mean of each no ATc replicate was set to 100% and the mean of each plus ATc replicate plotted. The means of each plus ATc replicate were compared to no ATc by a one-sample t-test. (C) Plaque assays of complemented lines, grown in the presence or absence of ATc for eight days. (D) Quantification of number of parasites per vacuole for parental and rSDH11 parasites. Parasites were grown in the presence or absence of ATc for 2 days before inoculation into fresh HFFs cells and grown for a further day in the presence or absence of ATc. Values are mean +/- S.D from three independent experiments. >100 vacuoles counted per replicate. ANOVA followed by Turkey's multiple pairwise comparisons was performed, and p-value from relevant pairs displayed. (E) Immunofluorescence assay of parental, cKD-ISU1-HA [35] (rISU1), rSDH10/SDHB-HA (rSDH10) and rSDH10/ SDHB-HA—SDH10-Ty (cSDH10) parasites treated with ATc for four days, labelled with GAP45 (magenta) to detect parasites and a lectin of Dolicos biflorus (DBL) (yellow) to outline nascent cyst walls. Scale bar is 5 µM. (E) Quantification of DBL-positive vacuoles from rSDH10 and rISU1 grown in the presence of ATC shown in *(D)*. Data are from three independent experiments; 100 vacuoles were counted per replicate. Values are mean -/+ S.D. compared by Student's t-test.

phenotype is also seen upon SDH disruption we analysed spontaneous differentiation to bradyzoites in culture upon depletion of SDH10. Parasites that have initiated differentiation were identified via an immunofluorescence assay, where cyst-like structures were identified by

staining with Dolichos biflorus lectin (DBL), which recognizes a glycoprotein in the nascent cyst wall. We found that 11% of the vacuoles were DBL-positive in the SDH10 mutant grown in the presence of ATc for four days suggesting that stage-conversion is triggered in this mutant (**Fig 4E**). No DBL-positive vacuoles were observed in the complemented line and fewer than 1% of vacuoles in the parental line grown in ATc were DBL-positive (**S4 Table**). We also saw DBL-positive vacuoles when we grew the rISU1, a mutant in mitochondrial Fe-S biogenesis, previously shown to show partial stage conversion [35]. Quantification of the number of DBL-positive vacuoles showed a higher number in the rISU line (55% vs 11%) (**Fig 4F and S4 Table**) suggesting the partial conversion phenotype is less pronounced in SDH mutants. The percentage of DBL-positive vacuoles in our mutant is also less pronounced than what was seen in other mutants with mitochondrial function defects such as parasites with a depleted subunit of complex III (30%) or of the mitoribosome (35%) [35], despite them being measured at the two-day timepoint, however, unlike the previous study which detected 5% of DBL positive vacuoles in parental lines treated with ATc we detected fewer than 1% (**S4 Table**), suggesting lower sensitivity in our hands, which might account for some of the observed difference.

## The new subunits are integral membrane proteins required to anchor SDH in the membrane

The finding that SDH10, 11, 15, 23 and 31 are essential for SDH formation and function, as well as their importance for parasite fitness, raised the question of their role within the complex. In the plant *Arabidopsis thaliana* mitochondrial SDH is also composed of more than four subunits (eight), and it was proposed that the non-catalytic subunits play a role in anchoring the complex in the membrane [36]. Similarly, SDH in the ciliate *Tetrahymena thermophila* includes fifteen subunits, and a structural study showed that the non-catalytic subunits mediate membrane anchoring [5]. Thus, we hypothesised that the same will be true for the subunits identified here. To test this hypothesis, we first performed structural prediction using Alpha-Fold [37,38], where we modelled the *Toxoplasma* SDHA and SDHB homologs along with SDH10,11,15,23 and 31 and superimposed them onto an experimentally determined eukaryotic SDH structure using the matchmaker tool on ChimeraX [39] (Avian SDH—PDB: 2H88 [40]). As expected SDHA and SDHB folding is mainly conserved albeit with a potential additional protrusion of SDHB compared with the Avian structure (**Figs 5A–5C and S5**). Furthermore, in line with our hypothesis, and with what has been proposed in other systems, the seven new components form a domain that is the right size, and in the correct position to mediate membrane anchoring, with 11 long helixes arranged perpendicular to SDHA and SDHB, and with each of the new subunits contributing some amino acids to the region potentially in contact with the membrane (**Fig 5C**). We also assessed the likelihood of these new SDH proteins containing a transmembrane domain using various prediction algorithms (**Table 1**). All were predicted to contain transmembrane domains, except SDH31, however there was a large degree of variability between prediction tools.

To provide support for the membrane-anchoring role of the newly confirmed subunits we performed sodium carbonate extractions, which allow discrimination between peripheral and integral membrane proteins [30]. Sodium carbonate extractions separate samples into a pellet fraction, containing integral membrane proteins, and a soluble fraction that contains both soluble and peripherally bound membrane proteins. We initially performed extractions on a line expressing a Ty tagged version of SDH15 in a line also expressing HA tagged SDHB. As expected SDHB, which is a peripherally associated protein in other systems and in our predicted structure, was in the soluble fraction with the peripherally associated membrane protein

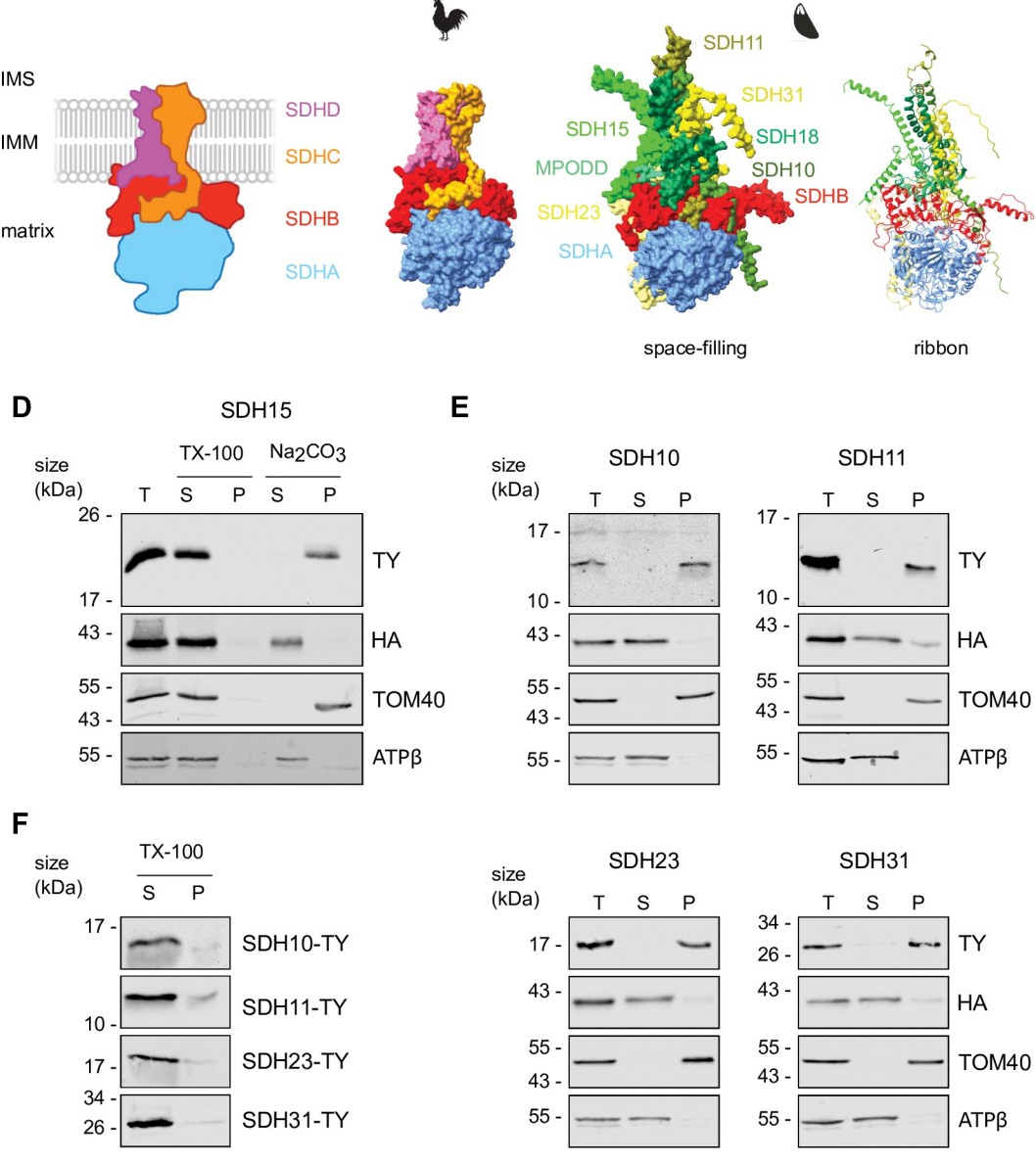

**Fig 5. Novel *Toxoplasma* SDH subunits are integral membrane proteins.** (A) Schematic of the canonical SDH, with the matrix facing SDHA and SDHB subunits in light blue and red, and integral membrane SDHC and SDHD subunits in orange and pink. (B) Diagram of Avian SDH crystal structure, (PDB: 2H88 [40]), colour-coded as in *A*. (C) Space-filled and ribbon homology models of *Toxoplasma* SDH, generated using alphafold and matchmaker. SDHA (blue) and SDHB (red) are similar to the canonical structure, and the new subunits are placed in the membrane region. (D) Immunoblot analysis of rSDH15/ SDHB-HA—SDH15-Ty parasites treated with 1% Triton X-100 (TX-100) or sodium carbonate ($Na_2CO_3$) at pH 11.5 and separated by centrifugation to give a supernatant (S) and pellet (P) fraction. A total fraction, before treatment (T) is also analysed. SDH15-Ty is found in the pellet fraction of $Na_2CO_3$, like the integral outer membrane protein TOM40, whereas SDHB-HA is found in the soluble supernatant fraction, like the matrix subunit of ATP synthase, ATPβ. (E) Immunoblot analysis of whole cell lysate extracted with sodium carbonate from cells expressing SDHB-HA and either SDH10/11/23 or SDH31-Ty. TOM40 and ATP β are used as marker proteins for the pellet and supernatant fraction respectively. (F) 1% Triton X-100 (TX-100) extractions and immunoblot analysis of SDH10/11/23 or SDH31-Ty.

ATP beta used as a control (**Fig 5D**). SDH15 was detected in the membrane fraction, along with the integral membrane protein TOM40 used as control (**Fig 5D**). Samples were also subjected to extraction with the detergent Triton X-100, to test that the proteins are not inherently insoluble. SDH15-Ty was detected in the soluble fraction, as expected. Similar results were obtained when these experiments were repeated with lines expressing Ty tagged subunits SDH10,11,23 and 31 in an SDHB-HA background (**Fig 5E and 5F**). This data suggests that the five new SDH subunits are integral membrane proteins, which is consistent with a role in anchoring the enzymatic domain into the membrane.

## A DY motif of SDH10 is required for SDH function and formation

In the commonly studied yeast and mammalian systems, the membrane anchoring domain of SDH contains the binding sites for ubiquinone. We reasoned that the new subunits would replace this critical function. However, our structural prediction did not reveal an identifiable binding site, and alignment of each of the new components with SDHC and SDHD homologs from different organisms failed to identify conserved motifs (**S6 Fig**). We therefore opted to focus on motifs that are strictly conserved among the myzozoan homologs of each of the five subunits. Sequence alignments performed for the myzozoan homologs of SDH11, 15, 23 and 31 did not identify amino acids that are both conserved among all homologs of a given subunit, and for which there is a proposed role in SDH from other organisms (**S7 Fig**). However, the alignment of myzozoan SDH10 homologs revealed a conserved DY motif (**Fig 6A**), the presence of which was also highlighted in the *Plasmodium falciparum* homolog, PF3D7_1448900, in a previous study [22]. A DY motif of SDHD was shown to contribute to one of the ubiquinone binding sites in mutational analysis of yeast SDH, whereby the tyrosine (Y) is forming a hydrogen bond with ubiquinone at the Qp site [41,42], and we thus hypothesised that it may play a similar role within SDH10. To test this hypothesis, we examined the effect of mutation in the DY motif on SDH activity, via complementation of rSDH10/ SDHB-HA with a minigene encoding a mutant form of SDH10-Ty where the DY motif is mutated to AA. First, we performed immunoblot analysis of the new line with parasites grown in the presence or absence of ATc and showed the SDH10$^{DY>AA}$-Ty protein is detected (**Fig 6B**). Notably, the protein levels of SDH10$^{DY>AA}$-Ty were increased upon addition of ATc, suggesting possible post-translational regulation of subunit abundance. Next, we confirmed that the SDH10$^{DY/AA}$-Ty is still localized to the mitochondrion (**Fig 6C**). Together these data suggests that the protein has likely not misfolded or degraded. Next, we performed the SDH enzymatic assay with the line complemented with the mutant subunit, rSDH10/SDHB-HA–SDH10$^{DY/AA}$-Ty. Complementation with SDH10$^{DY/AA}$-Ty was unable to restore SDH activity upon ATc treatment (**Fig 6D**). These observations suggest that the DY motif is required for SDH10 function. Finally, we tested the effect of the DY to AA mutation on complex formation. Native-PAGE and immuno-blotting revealed that the high-molecular weight band of SDHB-HA is reduced in the line complemented with SDH10$^{DY/AA}$-Ty at day three of ATc treatment (**Fig 6E**)**,** similar to what is seen when SDH10 is depleted. This observation suggests that the decrease in activity is accompanied with a decrease in complex formation, providing support for SDH10 fulfilling a role that replaces SDHD in yeast.

## Discussion

SDH is an important player in the mitochondrial metabolic network as the only enzyme shared by both the mETC and the TCA cycle. Numerous structural studies (for example [25,40,43,44]) demonstrated a remarkable conservation of the overall structure and four subunit composition of SDH in fungi and animals (opisthokonts) as well as in bacteria, and this

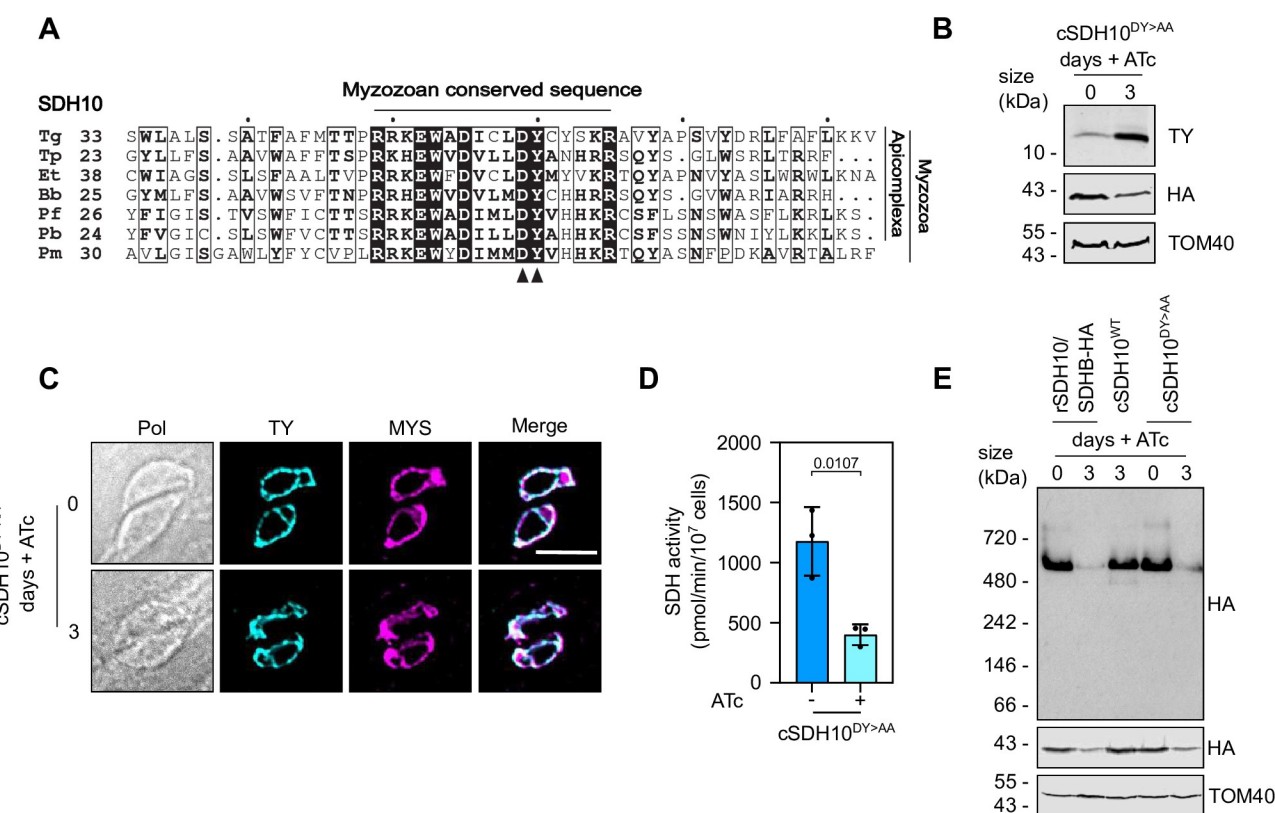

**Fig 6. DY motif of SDH10 is essential for SDH formation and activity.** (A) Alignment of SDH10 homologs from myzozoa, with conserved regions and the DY motif highlighted. *Tg Toxoplasma gondii; Tp, Theileria parva; Et, Eimeria tenella; Bb, Babesia bovis; PF, Plasmodium falciparum; Pb, Plamodium baeghei; Pm, Perkinsus marinus.*(B) Total lysate from rSDH10/ SDHB-HA—SDH10-DY>AA-Ty (cSDH10$^{DY>AA}$) grown in the absence (0) or presence of ATc for three days, separated by SDS-PAGE and immunoblot analysis performed with anti-Ty, anti-HA and anti-TOM40 antibodies. (C) Immunofluorescence assay analysis of rSDH10/ SDHB-HA—SDH10-DY>AA-Ty (cSDH10$^{DY>AA}$) grown in the presence or absence of ATc for three days and labelled with antibodies for Ty and the mitochondrial marker protein, MYS. Scale bar is 5 μM. (D) SDH activity from rSDH10/ SDHB-HA—SDH10-DY>AA-TY (cSDH10 DY>AA) grown in the absence (0) or presence of ATc for 3 days. Graphs show mean SDH activity -/+ S.D. from 3 independent experiments. No and plus ATc conditions were compared by a Student's t-test. (E) Total lysate from rSDH10/ SDHB-HA, rSDH10/ SDHB-HA—SDH10-Ty (cSDH10$^{WT}$) and rSDH10/ SDHB-HA—SDH10-DY>AA-Ty (cSDH10$^{DY>AA}$) grown in the absence (0) or presence of ATc for three days, separated by BN-PAGE and immunoblot analysis performed with anti-HA antibodies. Samples were also separated by SDS-PAGE and immunolabelled with anti-HA and anti-TOM40 antibodies.

initially pointed to a universally conserved enzyme [45]. However, studies in a growing repertoire of organisms, such as Embryophyta (land plants), Trypanosomatida (protozoan parasites) and Ciliophora (protists with cilia) revealed SDH complexes of varying compositions [4,5,36]. In many cases outside the opisthokonts and bacteria, a supramolecular complex, containing more than four subunits, have been described. In line with those discoveries, previous studies indicated that apicomplexan SDH consists of up to nine subunits which include orthologous of the catalytic SDHA and B along with up to seven non-catalytic subunits for which homologs are not found in organisms outside the Myzozoan clade [13,22]. Our work here validated five of the seven in *Toxoplasma gondii* SDH10, 11, 15, 23 and 31. We further demonstrated that each subunit is important for complex formation and function and therefore for parasite fitness. These findings add *Toxoplasma* to the growing repertoire of organisms that possess a supramolecular SDH.

What is the role of the divergent non-catalytic subunits found in different systems? One intuitive possibility, in light of the seeming universal conservation of the two catalytic subunits, is that any other subunit would contribute to forming the domain responsible for membrane

anchoring and ubiquinone binding. Finding from studies of the supramolecular SDH of plants support this option: in the four-subunit models, the membrane binding domain is constructed by the two transmembrane proteins SDHC-D/SDH3-4, which include three parallel membrane-spanning helices each, together forming a six-helix bundle. In plants, two of the non-catalytic subunits, SDH6 and 7, were shown to mediate membrane anchoring, and it was proposed that these two subunits provide missing helixes in SDH3 and 4 to generate a six-helix bundle like in the four subunit complexes [36]. In the case of *Toxoplasma*, our structural prediction points to 11 helixes contributed by the non-catalytic subunits which is different from the proposed six-helix model. However, in the predicted structure the helixes are stacked against one another unlike the "cage-like" bundle described in the four subunit structures. This is possibly a limitation of the prediction algorithm which direct structural studies may be able to address in the future. Nevertheless, the prediction provided support for a role in membrane anchoring for all five subunits studied here, and our experiments provide strong support to all five proteins being membrane bound, in agreement with this role. Moreover, depletion of each of the five subunits resulted in dissociation of SDHB from the complex, as seen by its native gel migration, providing further support that each of the five subunits is important for tethering of the catalytic subunits to the membrane complex.

The model whereby non-catalytic subunits form the membrane anchoring and substrate binding domain is further supported by our studied of SDH10. Our finding of a "DY motif" that is strictly conserved among all SDH10 myzozoan orthologs, along with its proposed role in ubiquinone binding in SDHD, pointed to the possibility that this subunit contributes to substrate binding. Our mutagenesis work then demonstrated a dependence of SDH structure and function on this DY motif. However, the interpretation of this latter finding is complicated due to an overall reduction of the fully formed SDH in this mutant, seen by native PAGE and immunoblot. The level of SDH10$^{DY>AA}$-Ty protein is increased compared to the wild-type version expressed from the same promoter. This elevation could be a result of post-translational feedback response to the defect in SDH assembly. Further research is required to reveal the mechanism behind this observation. Using sequence alignments and the structural prediction we were unable to pinpoint other amino acids beyond this DY motif that are likely to contribute to substrate binding. Potentially, future structural studies may reveal the identity of these amino acids and define both the ubiquinone binding site and the site of heme binding. Structural studies may have additional benefits as, considering the different composition of apicomplexan and human SDH complexes, there is a potential that the precise mechanism of binding may also be divergent and create an opportunity for the development of selective inhibitors for apicomplexan SDH. While we have not shown that SDH is strictly essential for fitness, its depletion has a clear fitness cost, and therefore parasite specific SDH inhibitors may make potential leads for drugs that can work in combination with other drugs.

While our data provide support to the divergent non-catalytic subunits fulfilling similar roles as SDHC and D, why so many subunits are needed to fulfil this role remains an open question. It is possible that some of these components, or part of them, mediate organism/lineage/clade specific functions in addition to the oxidation of succinate and reduction of ubiquinone. There is a growing number of examples of other complexes of the mitochondrial oxidative phosphorylation pathway that have additional functional modules in different lineages. For example, complex I of plants [46] and of different unicellular eukaryotes [47–49] include a carbonic anhydrase domain that is not found in complex I of bacteria and opisthokonts. Similarly, complex III of plants includes subunits that perform the activity of the mitochondrial processing peptidase (MPP), namely excising the transit sequences that mediates preproteins mitochondrial import [50,51]. Likewise, mitochondrial ATP synthase from different unicellular eukaryotes incorporate additional subunits that stabilize monomer

dimerization [52] and/or mediate higher order oligomerization [5,16,53]. Moreover, SDH was recently found in a respiratory super-complex for the first time, in the ciliate *Tetrahymena thermophila* [5,7]. From the SDH side, the interactions with the super-complex (specifically with complexes I and IV) are mediated by a 70-kDa module that is formed from species specific non-catalytic subunits. The new formation is proposed to induce membrane curvature that contributes to the shaping of the cristae and thus enable its bioenergetic functional specialization [5]. Thus, the SDH divergent non-catalytic components may mediate interactions that serve functions that are suitable to the specialised needs of the organism or clade. In support of this being the case also in *Toxoplasma*, our native migration and immunoblot experiments reveal complexes of higher-molecular weight than the ~500 kDa calculated by the complexome analysis but that still include SDHB. The identity of those complexes and of the subunits that mediate their formation remains to be explored.

Through the oxidoreduction of succinate and ubiquinone, SDH shuttles electrons into the mETC, where they are transferred to ultimately use oxygen as their acceptor. This electron transfer is required to maintain an electrochemical gradient through the coupled pumping of protons in complexes III and IV, which is necessary for ATP production by $F_oF_1$ATP synthase. Considering this essential role of the mETC, it is expected that SDH function will be important for parasite fitness. Depletion mutants could still form plaques in the host cell monolayer, albeit much smaller than the parental line, suggesting that despite the reduction of SDH function, the parasites could still grow. The parasite viability is in line with the observed residual respiration seen in our oxygen consumption assay. In *Toxoplasma*, four other dehydrogenases may act as a source of electrons for the chain. The matrix-facing single-subunit type II NADH dehydrogenase (NDH2) which retrieves electron from oxidation of NADH; dihydroorotate dehydrogenase (DHODH) which catalyses the rate limiting step in the pyrimidine synthesis pathway of oxidating dihydroorotate to orotate; a FAD-dependent glycerol 3-phosphate dehydrogenase (G3PDH); and a malate:quinone oxidoreductase (MQO), which participates in pyruvate metabolism. Thus, the observed residual respiration and growth could be attributed the activity of these dehydrogenases.

An emerging view regarding SDH is that its true biological function goes well beyond respiration [45] and other functions have been attributed to SDH subunits. For example, SDH subunits may participate in alternative assemblies, as demonstrated by yeast SDH3's interaction with the mitochondrial protein import machinery, which is necessary for the biogenesis of the protein import complex subunits Tim54 and Tim22, and for the import of precursor proteins via the TIM22 complex [54]. In support of this hypothetical scenario of a possible participation of SDH subunits in other complexes, we observed at least two of the subunits (SDH23 (**Fig 1B**), SDHB (**Fig 2A**,**2C and 2F**)) in additional high-molecular bands when analyzing their native gel migration. Future proteomic and structural analysis of SDH extracted using different detergents might provide insight into this intriguing possibility.

In summary, our work validated the composition of the supramolecular *Toxoplasma* SDH complex and demonstrated its importance for a fully functional mETC in these parasites and thus for their fitness. Our observations lead to new questions regarding the precise role of each of the components within the mechanism of activity of SDH as an oxidoreductase and potentially beyond that role.

## Materials and methods

### Cell culture

*Toxoplasma gondii* tachyzoites were cultured in human foreskin fibroblasts (HFF), sourced from ATCC (SCRC-1041). HFFs and parasites were culture in Dulbecco's Modified Eagle's

Medium (DMEM), containing 4.5 gL$^{-1}$ glucose, supplemented with 10% (v/v) fetal bovine serum, 4mM L-glutamine and penicillin/streptomycin antibiotics and grown at 37°C with 5% $CO_2$. When needed anhydrotetracycline (ATc) was added to the medium at a final concertation of 0.5 μM.

### Parasite genetic manipulation

CRISPR guided promoter replacement were performed in TATiΔKu80 cell line [32]. gRNAs were design around the start codon region of each gene using ChopChop tool (https://chopchop.cbu.uib.no/). Each gRNA (**S1 Table**) was cloned into a vector containing the U6 promoter and expressing CAS9-GFP (Tub-Cas9-YFP-pU6-ccdB-tracrRNA) using the BsaI restriction site [32,55]. A DNA midi-prep (Qiagen) was used to purify the final plasmid, per the manufacturer's protocol. The pDTS4myc plasmid [32] was used as a template for amplification of DHFR selectable cassette and ATc repressible promoter by PCR. gRNAs and cassette integration were inserted in TATiΔKu80/SDHB-HA [13] cell line by electroporation. Cassette integration was selected with pyrimethamine (1 μM) and cloned by serial dilution. Positive clones were tested by PCR using the primers from **S1 Table**.

For the complementation of the promoter replacement cell lines, cDNAs were cloned (primers in **S1 Table**) into a pTUB8mycGFPMyoATy expression vector via EcoRI, MfeI and NsiI restriction sites [56]. The vectors were electroporated into the respective promoter replacement lines and integration selected with mycophenolic acid (25 mg/mL) and xanthine (50 mg/mL). The mutation of the DY motif to AA of rSDH10/SDHB-HA was performed in the pTUB8-SDH10-Ty vector (generated above) with the primers 227920_DY>AA R and F (**S1 Table**) using the commercial QuikChange II Site-Directed Mutagenesis Kit (Stratagen) according to manufacturers' instructions.

### qRT-PCR

To measure the levels of gene downregulation, qRT-PCRs were performed in rSDH10,11,15,23,31/SDHB-HA cell lines grown in the absence or presence of ATc for 2 days. Parasites were harvested and host cell debris were removed by a 3 μm polycarbonate filter. RNA was extracted using RNeasy kit (Qiagen) with an added DNAseI step. cDNA was made with high-capacity RNA-to-cDNA kit (AppliedBiosystems). qRT-PCRs were set up with PowerSYBR green master mix (ThermoFisher) using 10 ng of cDNA as a template. Specific primers for each gene and actin were used (**S1 Table**). Twenty-five cycles were performed using 7500 Real Time PCR System (Applied Biosystems, Waltham, MA, USA). To calculate the relative expression of samples treated or not with ATC, the double Δ Ct method [13] was used with actin mRNA as an internal control. Three independent experiments were performed. The software GraphPad Prism 9.2.0 was used for data plotting.

### Immunofluorescence assays

To assess subunit localization, parasites were transfected with pTUB8-subunit-Ty expression vector outlined above and inoculated onto HFFs grown on glass coverslips. After 1 day cells were fixed with 4% paraformaldehyde. Cells were labelled with primary antibodies against Ty (1:800, anti-mouse, [57]) and the mitochondrial protein MYS (1:1000, anti-rabbit,[29]), before being labelled with secondary antibodies (Alexa Fluor Goat anti-mouse 488 A11029, 1:1000 and Alexa Fluor Goat anti-Rabbit 568 A11036, 1:1000). Slides were imaged on a DeltaVision Core microscope (Applied Precision) using the 100x objective and z-stacking. Images were deconvolved using SoftWoRx software and processed using FIJI software.

Super resolution microscopy: slides were imaged using Elyra PS.1 super-resolution microscope (Zeiss, Jena, Germany) with a 63×/1.4 NA oil-immersion objective using ZEN black software (Zeiss). Vacuoles with 4 tachyzoites were captured with depth in 0.091μm increments. Images were acquired with 3 phase settings and structure illuminated and aligned in the imaging software. The 3D models were reconstructed in Imaris software (Oxford Instruments).

Expansion microscopy: Ultrastructure Expansion Microscopy (U-ExM) was performed according to [58] with some modifications: the parasites were grown in an HFF monolayer on glass coverslip in 24-well plates. The gels were denatured at 85˚C for 90 min. Primary antibodies used were rat anti-HA (1:250, Sigma, Merck, Gillingham, UK), mouse anti-Ty (1:250, [57]) and rabbit anti-TOM40 (1:250, [30]). Secondary antibodies used in this study were Alexa Fluor Goat anti-mouse 488 A11029, 1:250, Alexa Fluor Goat anti-Rat 594. 1:250 and Alexa Fluor 488-conjugated goat anti-rabbit-IgG, 1:250. Images were acquired using Elyra PS.1 super-resolution microscope (Zeiss, Jena, Germany) with a 63×/1.4 NA oil-immersion objective using ZEN black software (Zeiss). Vacuoles with two tachyzoites were captured with depth in 0.091μm increments. Images were acquired with 3 phase settings and structure illuminated and aligned in the imaging software. Images were processed and distance measured in FIJI. The 3D models were reconstructed in Imaris software (Oxford Instruments). Expansion factor was 3x and was determined after the first round of expansion. The distances measured were divided by the expansion factor.

## Extracellular flux analysis

Oxygen consumption rate (OCR) and extracellular acidification rate (ECAR) was measured using a Seahorse XF HS Mini Analyser (Agilent technologies). The method was adapted from [34] for compatibility with the 8-well plate format. Briefly, parasites were incubated in the presence or absence of ATc for three days before harvest. Fully lysed parasites were passed through a 3 μm polycarbonate filter, washed and resuspended with Seahorse XF DMEM base medium supplemented with 5 mM glucose and 1 mM glutamine, and the $1.5 \times 10^6$ were added per well to a plate treated with poly-L-lysine. Parasite OCR and ECAR were then measured using the Seahorse XF HS Mini Analyser. Background, non-mitochondrial OCR was taken as the OCR measurement after addition of 1 μM atovaquone. Basal and maximal mitochondrial OCR were calculated by taking the OCR measurement before or after addition of 1 μM FCCP respectively and subtracting the background rate. Each experimental run was performed with a single line grown in the absence or presence of ATc with three technical replicates. three independent experiments were performed for each line.

## Growth analysis

For plaque assays, monolayers of HFFs cells in 6-well plates were infected with 100–200 tachyzoites and grown in the presence and absence of ATc for 8 days. Cells were fixed with methanol and stained with a 0.4% crystal violet solution. 10 plaques per condition were measured from three biological replicates using Image J.

For cyst quantification the respective cell lines, rSDH10/SDHB-HA, rSDH10/SDHB-HA-SDH10-Ty, parental and cKD-TgISU1-HA [35] were grown in the presence or absence of ATc for 4 days, on HFFs grown on glass coverslips, followed by an IFA. Parasites were stained with the GAP45 antibody (1:1000, [59]) and a lectin of Dolicos biflorus (DBL) marker that labels nascent cyst walls (1:500, FL-1031, Vector laboratories). To quantify the percentage of positive cysts, 100 vacuoles were counted from three biological replicates.

Replication assays were performed by growing parental and rSDH11/SDHB-HA parasites in the absence or presence of ATc for two days and then infecting confluent HFFs for 24 hours

in the same conditions. Cells were washed with PBS to remove extracellular parasites and fixed with 4% PFA. Immunofluorescence assays were performed as described above using the GAP45 antibody. The number of vacuoles containing one, two, four or eight+ parasites were counted for 100 vacuoles. Three independent experiments were performed.

## SDS-PAGE and immunoblot analysis

Parasite samples were resuspended in $1 \times$ NuPAGE LDS loading dye (Invitrogen, Paisley, UK) supplemented with 5% *v/v* beta-mercaptoethanol or Laemmli buffer (2% (w/v) SDS, 125 mm Tris–HCl pH 6.8, 10% (w/v) glycerol, 0.04% (v/v) β-mercaptoethanol, and 0.002% (w/v) bromophenol blue), heated at 95˚C for 5 min and were separated on a SDS-PAGE gel, using a EZ-Run Prestained Rec protein ladder as a molecular weight marker. Proteins were transferred under semi-dry conditions in Towbin buffer (0.025 M Tris 0.192 M Glycine 10% Methanol) onto nitrocellulose membrane (0.45 μm Protran, Merck, Gillingham, UK). Membranes were then labelled with the relevant primary antibodies: anti-HA (1:500, rat, Sigma, Merck, Gillingham, UK), anti-TOM40 (1:2000, rabbit, [30]), anti-Ty (1:1000, mouse, [57]) and anti-ATPβ (1:2000, rabbit, Agrisera AS05 085) before being labelled with secondary fluorescent antibodies IRDye 800CW and IRDye 680RD (1:10,000, LIC-COR, Lincoln, NE, USA) and visualized with Odyssey CLx imaging system.

## Immunoprecipitations

For co-immunoprecipitation and immunoblot analysis: $9 \times 10^7$ rSDHB15/ SDHB-HA +-SDH15-Ty parasites were split into three aliquots and lysed in 1% digitonin, 50 mM Tris-HCL pH 7.4, 150 mM NaCl, 2 mM EDTA. One aliquot was retained as an input sample while the other two were cleared by centrifugation at 18,000 x g and the supernatant incubated with either anti-HA or anti-Ty agarose beads overnight at 4˚C. The agarose beads were pelleted and the supernatant was TCA precipitated to give the "unbound" fraction. The beads were then washed in lysis buffer with 0.1% digitonin and resuspended in laemmli buffer to give the "bound" fraction. All fractions were then subjected to immunoblot analysis, as above.

For identification by mass spectrometry: $5 \times 10^8$ cells from either SDHB-HA or QCR2-HA [13] were lysed in a 1% βDDM lysis buffer (50 mM Tris-HCL pH 7.4, 150 mM NaCl, 2 mM EDTA) and immunoprecipitated with anti-HA agarose beads (Pierce). Beads were washed twice with buffer containing 0.05% βDDM. Bound proteins were eluted with 50 mM NaoH. Total, unbound and elute fractions were taken for immunoblot analysis. Elute corresponding to ~$1.2 \times 10^8$ cells was sent for analysis by mass spectrometry.

## Mass spectrometry

Sample processing: Samples resolved for 2 cm via SDS-PAGE using a 4–12% gel. After staining with Quick Coomassie Stain (Generon) the 2 cm lanes were excised and diced. The gel pieces were then alkylated with iodoacetamide (Sigma-Aldrich) before being subjected to in-gel digestion using 1 mg/ml Trypsin (Thermo Scientific) at a final concentration of 12.5 μg/ml. Digested peptides were then dried down using vacuum centrifugation and resuspended to 50 μl with 1% formic acid (Fisher Chemical).

LC-MS analysis: LC-MS analysis was performed by the FingerPrints Proteomics Facility (University of Dundee). Analysis of peptide readout was performed on a Q Exactive plus, Mass Spectrometer (Thermo Scientific) coupled with a Dionex Ultimate 3000 RS (Thermo Scientific). LC buffers used are the following: buffer A (0.1% formic acid in Milli-Q water (v/v)) and buffer B (80% acetonitrile and 0.1% formic acid in Milli-Q water (v/v). Reconstituted samples, in formic acid, were loaded in aliquots of 15 μl at 10μL/min onto a trap column (100 μm × 2

cm, PepMap nanoViper C18 column, 5 μm, 100 Å, Thermo Scientific) which was equilibrated with 0.1% Trifluoroacetic acid. The trap column was washed for 3 min at the same flow rate and then the trap column was switched in-line with a Thermo Scientific, resolving C18 column (75 μm × 50 cm, PepMap RSLC C18 column, 2 μm, 100 Å) which was equilibrated in 2% buffer B for 17 min. The peptides were eluted from the column at a constant flow rate of 300 nl/min with a linear gradient from 2% buffer B to 5% buffer B in 3min, 5% buffer B to 35% buffer B in 64 min, and then from 35% buffer B to 98% buffer B in 2 min. The column was then washed with 98% buffer B for 15 min. Two blanks were run between each sample to reduce carry over. The column was kept at a constant temperature of 50˚C. Q-exactive plus was operated in data dependent positive ionization mode. The source voltage was set to 3.0 kV and the capillary temperature was 250˚C.

A scan cycle comprised MS1 scan (m/z range from 350–1600, ion injection time of 20 ms, resolution 70 000 and automatic gain control (AGC) 1x106) acquired in profile mode, followed by 15 sequential dependent MS2 scans (resolution 17500) of the most intense ions fulfilling predefined selection criteria (AGC 2 x 103, maximum ion injection time 100 ms, isolation window of 1.4 m/z, fixed first mass of 100 m/z, spectrum data type: centroid, intensity threshold 2 x 104, exclusion of unassigned, singly and >5 charged precursors, peptide match preferred, exclude isotopes on, dynamic exclusion time 45s). The HCD collision energy was set to 27% of the normalized collision energy. Mass accuracy was checked before the start of samples analysis.

Data analysis: Label-free analysis was performed in MaxQuant version 2.0.3.0 using the RAW files generated with data manipulation in Microsoft Excel Office 365 to provide comparisons between control and treated samples. Further data analysis was performed on proteins detected in all four independent experiments for each sample using Perseus version 1.6.15.0 to generate volcano plot data and perform student t-tests. Final volcano plot figures were generated using GraphPad Prism 8.4.3.

The full data in deposited in PRIDE database. Project Name: Immunoprecipitation of *Toxoplasma gondii* succinate dehydrogenase complex. Project accession: PXD047027.

## Blue and clear native PAGE

Blue native page (BN-PAGE) was performed as described previously [13] Briefly, parasites were resuspended in solubilization buffer (750 mM aminocaproic acid, 0.5 mM EDTA, 50 mM Bis-Tris-HCl ph 7.0, 1% (w/v) n-dodecylmaltoside), incubated for 10 minutes on ice and then centrifuged at 18,000 x g at 4˚C for 30 minutes. The resulting supernatant was combined with Coomassie G250 (NativePAGE) to a final concentration of 0.25% detergent and 0.0625% Coomassie G250. Samples were separated on a Native PAGE 4–16% Bis-Tris gel using Native-Mark (ThermoFisher) as a molecular weight marker. The gel was run at 80 V for 60 minutes using a dark cathode buffer (5% of blue cathode buffer (Invitrogen, BN2002) and 5% of 20x Running buffer (Invitrogen, BN2001) followed by 90 minutes at 250 V in a light cathode buffer (0.5% of blue cathode buffer (Invitrogen, BN2002) and 5% of 20x Running buffer (Invitrogen, BN2001). Proteins were transferred onto a PVDF membrane (0.45 μm, Hybond, Merck) via wet transfer in Towbin buffer (0.025 M TRIS 0.192 M Glycine 10% Methanol) for 60 minutes at 100 V. Immunolabelling was carried out with appropriate primary antibodies: anti-HA (1:1000, rat, sigma), anti-TOM40 (1:2000, rabbit, [30]), and anti-Ty (1:800, [57]) and detected using anti-rat (ab6845, abcam), anti-rabbit (1:10000, W4011, Promega) and anti-mouse (1:10000, W4021, Promega) secondary horseradish peroxidase-conjugated antibodies. Detection was through chemiluminescence using Pierce ECL Western Blotting Substrate and an x-ray film.

Clear native Page (CN-Page) followed by complex IV activity assay were performed according to [60]. Parasites were resuspended in solubilization buffer (50 mM NaCl, 2 mM 6-amino-hexanoic acid, 50 mM imidazole, 2% (w/v) n-dodecylmaltoside, 1 mM EDTA–HCl pH 7.0) and incubated in ice for 10 min, centrifuged at 18,000 x g at 4°C for 15 minutes. Supernatants were then mixed with glycerol and ponceau S to a final concentration of 6.25% and 0.125% respectively. Samples were separated on a pre-cast Native PAGE 4–16% Bis-Tris gel using NativeMark (ThermoFisher) as a molecular weight marker. Complex IV oxidation activity was shown by incubating gels in a 50 mM $KH_2PO_4$, pH 7.2, 1 mg ml−1 cytochrome *c*, 0.1% (w/v) 3,3′-diaminobenzidine tetrahydrochloride solution until brown precipitates were visible.

### Enzymatic activity assay

Succinate dehydrogenase (SDH) activity was measured using a spectrophotometric assay adapted from [61]. *T. gondii* parasites were grown in the absence and presence of ATc for 3 days, before extracellular parasites were harvested, and host cell debris removed with a 3 μm polycarbonate filter. $3 \times 10^7$ parasites resuspended in 100 μl of mitochondrial assay solution lysis buffer (70 mM sucrose, 220 mM mannitol, 10 mM $KH_2PO_4$, 5 mM $MgCl_2$, 2 mM HEPES, 1 mM EGTA, 0.2% w/v bovine serum albumin (BSA), and 0.2% w/v digitonin, pH 7.2) and incubated for 1 hour at 4°C. The resulting parasite lysate was then added to an assay buffer (final volume 1 mL) containing 20 mM succinate, 25 mM $KH_2PO_4$, 0.002175% w/v DCPIP, 1 μM atovaquone, 2.5 mg/mL BSA and 40μM decyclubiquinone (DUB) and absorbance at 600 nM measured every 2 minutes, for 40 minutes. Lysate from equal starting number of parasites was used for each replicate. MQO activity was measured using the same protocol, except using 20 mM malate instead of succinate. Enzymatic activity was calculated using an extinction coefficient of DCPIP of 19.1 $mM^{-1}$ $cm^{-1}$.

### Sodium carbonate extractions

To separate integral membrane proteins from peripheral membrane proteins, parasites were treated with 100 mM $Na_2CO_3$ and incubated at 4°C for 2 hours. The pellet and supernatant fraction were separated by ultracentrifugation at 189,000 x g. The supernatant fraction was TCA precipitated before it and the pellets fraction were resuspended in laemmli buffer. To test the solubility of proteins, parasites were resuspended in 1% triton X-100, incubated at 4°C for 2 hours, and pellet and supernatant fractions separated by centrifugation at 16,000 x g. Fractions, as well as a total parasite sample, were then tested by immunoblot analysis.

### Structural predications

The predicted structures of each *T. gondii* SDH subunit were obtained from the AlphaFold database [37,38]. The mitochondrial targeting sequences were predicted using MitoProt [62] and removed. Truncated sequences were then aligned against an avian complex II structure (PDB 2H88) using the matchmaker tool in UCSF ChimeraX [39].

### Supporting information

**S1 Fig.** (A) Super resolution microscopy images shown as Z-stack (Left and middle panel) and 3D reconstruction (right panel) of SDHB-HA and SDH15-Ty, with the mitochondrial marker protein TOM40. Scale bar is 2 μM. (B) Ultra-expansion microscopy images from Fig 1B. Left: Merge of SDHB-HA/SDH15-Ty (orange) and TOM40 (pink) with inset area highlighted, Right: close-up of inset with white line indicating area of intensity calculation. Scale bar is 1 μM (C) Intensity plot of SDHB-HA/SDH15-Ty and TOM40 signal from part *B*.

(D) Full immunoblot of the data shown in Fig 1C.
(TIF)

**S2 Fig.** (A) Immunoblot analysis of whole cell lysate extracted from cells expressing SDH10, SDH11, SDH15, SDH23 or SDH31-Ty performed with anti-Ty antibodies. (B) Schematic of the promoter replacement strategy allowing knock-down of a gene of interest (GOI) with the addition of anhydrotetracycline (ATc). (i) CRISPR/CAS9 guided cut at the predicted promoter/ATG boundary, (ii) a repair cassette containing the ATc repressible promoter (T7S4), the dihydrofolate reductase (DHFR) selection marker, and homology to the promoter/ATG boundary, is inserted between the promoter and GOI during cut-repair guided by the homology sequences, (iii) GOI, under the control of the ATc repressible promoter, is down regulated when ATc is added. The black arrows represent the primers used to confirm integration via PCR (primers in **S1 Table**). (C) PCR analysis of rSDHB10,11,15,23 or 31, using the primers in *B*, to confirm the integration of the DHFR and repressible promoter. (D) qRT-PCR analysis of relative transcript levels of SDH10,11,15,23 or 31 in the respective promoter replacement lines after growth in ATc for two days, No ATc was set to one and plus ATc conditions were compared by a one-sample t-test. Bars represent the mean ± S.D. (n = 3).
(TIF)

**S3 Fig.** Immunoblot analysis of whole cell lysate extracted from SDHB-HA (A) or QCR2-HA (B) and immunoprecipitated with anti-HA beads, to produce total lysate, unbound and elute fractions. Samples were separated by SDS-PAGE, blotted, and detected using anti-HA antibody to label immunoprecipitated proteins, and anti-TOM40 as an unrelated mitochondrial protein control.
(TIF)

**S4 Fig. Immunofluorescence assay analysis of parental, rSDH15 and rVDAC [33] parasites grown in the absence or presence of ATc for three days.** Mitochondrial morphology is visualised with antibodies against the mitochondrial marker protein TOM40. Scale bar is 5 μM.
(TIF)

**S5 Fig.** (A) ribbon homology models of *Toxoplasma* SDH, front and back view, generated using alphafold and matchmaker. Individual subunits colour coded. (B) Homology model of subunits SDHA and SDHB (C) Homology models of the novel SDH subunits.
(TIF)

**S6 Fig. Alignment of MPOD, SDH10, SDH11,15,23,31 with SDHC and SDHD from species with a canonical SDH complex.** *Gg, Gallus gallus*; *Bt, Bos taurus*; *Hs, Homo sapiens*; *Dm, Drosophila melanogaster*; *Bs, Bradyrhizobium species*; *Bj, Bradyrhizobium japonicum*; *Ra, Reclinomonas americana*; *Rp, Rickettsia prowazekii*; *As, Ascaris suum*; *Ps, Paracoccus species*; *Pd, Paracoccus denitrificans*; *Rr, Rhodospirillum rubrum*; *Ec, Escherichia coli*; *Sc, Saccharomyces cerevisiae*; *Pp, Porphyra purpurea*.
(PDF)

**S7 Fig. Alignment of SDH11,15,23,31 homologs from myzozoa.** *Tg Toxoplasma gondii*; *Tp, Theileria parva*; *Et, Eimeria tenella*; *Bb, Babesia bovis*; *PF, Plasmodium falciparum*; *Pb, Plamodium baeghei*; *Pm, Perkinsus marinu; Po, Perkinsus olseni*.
(TIF)

**S1 Table. Primer table: summary of primers used in this study.**
(XLSX)

**S2 Table. Mass spectrometry data from SDHB and QCR2 immunoprecipitations.**
(XLSX)

**S3 Table. SDH and MQO enzymatic activity measurements.**
(XLSX)

**S4 Table. Quantification of DBL-positive vacuoles in parental, rISU1, rSDH10 and cSDH10 parasites.**
(XLSX)

## Acknowledgments

We thank Leandro Lemgruber from the imaging facilities of the Wellcome Centre for Integrative Parasitology for support for the microscopy work. We thank Giel van Dooren and Soraya Zwahlen of the Australian National University for sharing their protocol for complex II activity assay using *T. gondii* cell lysate. We thank EupathDB and ToxoDB for providing useful and free access to genome databases. We would like to acknowledge Alan Score of the FingerPrints Proteomics Facility at the University of Dundee, which is supported by the 'Wellcome Trust Technology Platform' award [097945/B/11/Z].

## Author Contributions

**Conceptualization:** Andrew E. Maclean, Lilach Sheiner.

**Data curation:** Mariana F. Silva.

**Formal analysis:** Kiera Douglas, Sofia Sandalli.

**Funding acquisition:** Andrew E. Maclean, Lilach Sheiner.

**Investigation:** Mariana F. Silva, Kiera Douglas, Sofia Sandalli, Andrew E. Maclean, Lilach Sheiner.

**Methodology:** Mariana F. Silva.

**Supervision:** Andrew E. Maclean, Lilach Sheiner.

**Validation:** Mariana F. Silva.

**Visualization:** Mariana F. Silva.

**Writing – original draft:** Andrew E. Maclean, Lilach Sheiner.

**Writing – review & editing:** Mariana F. Silva, Andrew E. Maclean, Lilach Sheiner.

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
