## [Decision Letter · Decision Letter 0]

2 Aug 2023

Dear Dr. Sheiner,

Thank you very much for submitting your manuscript "Functional and biochemical characterisation of the Toxoplasma gondii succinate dehydrogenase complex" for consideration at PLOS Pathogens. As with all papers reviewed by the journal, your manuscript was reviewed by members of the editorial board and by several independent reviewers. The reviewers appreciated the attention to an important topic. Based on the reviews, we are likely to accept this manuscript for publication, providing that you modify the manuscript according to the review recommendations.

As you will see, both reviewers were overall positive about the manuscript. Please pay close attention to the in vitro experiments requested, as they seem reasonable to support the central claims made in the paper. I would also agree with Reviewer #2 that mouse experiments are required to support any claims of an in vivo requirement for something not essential in vitro. Please address this concern either experimentally or by modifying the manuscript.

Sincerely,

Michael L Reese, PhD

Academic Editor

PLOS Pathogens

Margaret Phillips

Section Editor

PLOS Pathogens

Kasturi Haldar

Editor-in-Chief

PLOS Pathogens

orcid.org/0000-0001-5065-158X

Michael Malim

Editor-in-Chief

PLOS Pathogens

orcid.org/0000-0002-7699-2064

As you will see, both reviewers were overall positive about the manuscript. Please pay close attention to the in vitro experiments requested, as they seem reasonable to support the central claims made in the paper. I would also agree with Reviewer #2 that mouse experiments are required to support any claims of an in vivo requirement for something not essential in vitro. Please address this concern either experimentally or by modifying the manuscript.

Reviewer Comments (if any, and for reference):

Reviewer's Responses to Questions

**Part I - Summary**

Reviewer #1: The authors test the localization, essentiality, and function of several new, parasite-specific proteins identified in a prior complexome profile of ETC complex II in Toxoplasma gondii. Their prior study had identified two of the four canonical members of SDH known in mammals but suggested seven novel subunits in parasites. This study confirms five of the seven proteins as core members of complex II and provides direct evidence that complex II function is critical for parasites.

This strong study is well executed and features good controls. The results and conclusions impactfully advance understanding of the divergent composition of apicomplexan ETC complex II. There are a few inconsistencies and questions noted below, answering which will tighten and strengthen the major conclusions and implications.

Reviewer #2: In the reported work, Marina Silva et al verified that several novel subunits of the purported mitochondrial complex II previously identified in complexomic studies are indeed components of a large mitochondrial complex and are required for succinate-ubiquinone oxidoreductase activity in the parasite, as expected for complex II components. The experiments addressing these points appear to be thorough and the results compelling. Statements by the authors regarding the suitability of SDH/complex II as a drug target in Toxoplasma and other apicomplexans are less convincing since the subunits proved not strictly necessary for growth in vitro. Thus the study should be bolstered by in vivo experiments with knockdown parasite lines.

**Part II – Major Issues: Key Experiments Required for Acceptance**

Reviewer #1: 1. Lines 143-146: The gels in Fig. 1C are not convincing that all 4 proteins migrate in the same-sized complex ~500 kDa, since the bands for 10 and 31 appear to be ~700 based on the markers. Since the MW sum of all 9 proposed SDH proteins is ~230 kDa (Table 1), the authors are presumably detecting a higher-order oligomeric complex, but this aspect is not discussed. One explanation for the differing BN-PAGE band sizes could be that these proteins are present in distinct oligomeric states for SDH. Idiosyncrasy and gel-to-gel variability are also normal in BN-PAGE experiments. To better resolve and understand these differences, can the authors run lysates from the four tagged lines on the same BN-PAGE gel (and/or co-load lysates from different tags in the same lane) to directly establish if the four proteins are in dominant complexes of similar or distinct sizes?

2. The structural homology modeling in Fig. 5C is interesting and insightful. It would be very helpful if the authors could label the individual parasite proteins modeled in this panel.

3. For Fig. 6D, does the DY-AA mutation destabilize SDH10 and thus result in diminished levels of this protein (that are insufficient to stabilize complex II upon knockdown of endogenous SDH10?) The authors can detect the mutant protein in the mitochondrion but this IFA does not enable evaluation of relative protein levels. Probing for TY in the Fig. 6D blot would be sufficient to assess.

Reviewer #2: An important part of the claimed significance of this work is the potential of the SDH complex to be a drug target in Toxoplasma and other apicomplexan parasites. This should be supported by experiments in animals since the complex was found not to be absolutely required for propagation in vitro.

**Part III – Minor Issues: Editorial and Data Presentation Modifications**

Reviewer #1: 1. Lines 37-38: should be “starts” and “studies”

2. Line 63: should be “have”

3. Lines 63-69: molecular divergence by itself doesn’t seem the most compelling rationale for studying these complexes in Apicomplexa. Rather, the implications and insights of this divergence for understanding pathogen evolution and identifying pathogen-specific metabolic vulnerabilities distinct from the host seem to be the major motivators for defining and understanding these differences from mammals (and yeast)

4. Lines 74-79: These statements are overly general, since complex II/SDH (Pubmed 25843709 and 22628552) and ATP synthase (Pubmed 25831536) are not essential for blood-stage Plasmodium.

5. Does complex II or any of the new membrane-bound subunits bind heme? Does the BN-PAGE band ~500 kDa give a positive “heme stain” by luminol-based chemiluminescence (e.g., Pubmed 12672416)? The role of heme in SDH in mammalian and yeast complex II remains uncertain but appears to contribute to complex stability. Given the paucity of heme-dependent proteins in apicomplexan parasites relative to mammals, understanding of whether T gondii SDH retains or lacks bound heme would further clarify molecular similarity or differences from opisthokonts.

6. Lines 134-136: Based on current Fig. 1B, it is not obvious that SDHB and TOM40 have resolvably distinct localization within mitochondria. Higher magnification and an intensity vs distance plot along a line orthogonal to the membranes showing discrete intensity maxima would be needed to convincingly make this conclusion.

7. Can the authors provide the full mass spec data for pull-downs of SDHA, SDHB, and cyt c1 to better evaluate interaction specificity? Table S2 is presumably an over-simplification of interactors, and pull-down with many more proteins was observed in all cases.

8. Lines 190-191: reduction in SDHB-HA levels upon knockdown of other subunits seems to correlate with loss of the ~500 kDa band, which suggests that SDHB stability depends on complex II integrity. The authors may wish to explicitly discuss this connection.

9. Line 601: the authors presumably mean “/mM /cm” for the extinction coefficient units.

10. Line 209: the authors describe use of parasite lysates here but the methods (lines 590-601) only mention intact parasites. Can the authors check and clarify what sample they used and how they normalized protein concentrations across samples?

11. Line 852: Does “three independent experiments” mean technical (same samples run three times) or biological (three different samples derived from different parasite knockdown experiments) replicates?

12. Lines 93 and 243: Apicomplexa like Plasmodium lack canonical multi-subunit complex I but express a single-subunit type II NADH dehydrogenase that is expected to generate reduced ubiquinol which contributes electrons to the ETC (e.g., Pubmed 30964863). The authors’ statements here seem a bit misleading.

13. Lines 249-250 and 421-423: An alternative explanation is that SDH is essential for Toxoplasma but the knockdowns of SDH subunits were insufficiently stringent to fully ablate SDH activity, resulting in residual parasite growth. More detailed experiments would be needed to differentiate essential versus fitness conferring (but not strictly essential) contributions. The authors may wish to acknowledge this possibility, especially as protein levels were not examined after ATc addition.

14. Fig. 4F: The authors should include a “no ATc” or parental control to establish baseline differentiation for comparison to the two experimental knockdowns.

15. Opisthokonts appears to be misspelled as “opistokonts” in multiple locations (lines 352,355, 402).

16. Are SDH knockdown parasites (tachyzoites or bradyzoites) more sensitive to atovaquone (additive or synergistic)? Atovaquone monotherapy is associated with partial efficacy, which limits its utility for treating chronic or acute toxoplasmosis. Would combinatorial inhibition of SDH and atovaquone treatment prove beneficial for more stringently killing of parasites and preventing recrudescence? This experiment is not required for the authors’ conclusions but would substantially increase the translational relevance of the present study.

Reviewer #2: Lines 74-75: "essential throughout their complicated life cycles" doesn't apply to all apicomplexans, i.e. Cryptosporidium, but also P. berghei (manuscript ref 11).

line 78: "dual target" inhibitors of Plasmodium may have been explored, but the lethality is almost certainly due to inhibition of complex III since SDH is clearly dispensable in the blood stage (similar to the case recently reported for NADH dehydrogenase inhibitors that also inhibit complex III in P. falciparum).

line 93-94 and line 243: "in the absence of mETC complex I, apicomplexan SDH is one of the first points of entry for electrons..." The absence of complex I is irrelevant since there are NADH dehydrogenases, as well as other quinone-dependent dehydrogenases, that all donate electrons at the same entry point as SDH.

line 360 and Table I: "study by Zwahlen et al; provided validation for the additional two, SDH18 and MPODD" Zwahlen et al is an important reference but is not listed in the reference section.

line 366: "substrate" Since there are multiple substrates, specify that it is ubiquinone.

line 378: "validated that all five proteins are membrane bound," "Validated" seems to be too strong a conclusion, since some may be bound indirectly via binding to another integral membrane subunit, which is suggested by the subunits predicted to lack TM segments (Table I: SDHB, SDH15, SDH31, MPODD).

line 386: "demonstrated a dependence of SDH function on this DY motif." Should be "structure or function", i.e., a trivial reason for lack of function is failure to fold correctly. This is similar to the qualifications stated by the authors in the following sentence, but even is a mutated protein is correctly localized, there may be conformational changes that block function.

line 421: "it is expected that SDH function will be essential for the parasites" I wouldn't expect it to be essential since there are several other dehydrogenases that donate electrons to ubiquinone (and it isn't essential in Plasmodium).

Fig. 3A: Next to last bar is mislabeled.

Fig. 5: suggest moderately enlarging B and C, and color-coding the gray subunits so the individual subunits having membrane segments can be identified and compared to predictions in Table I.

Table I: One of the TMD predictors gives predictions that mostly differ from the other two and happens to not be peer reviewed. Suggest substituting results from a peer reviewed algorithm. One possibility that combines the predictions of multiple predictors is CCTOP [László Dobson, István Reményi and Gábor E. Tusnády (2015)

Nucleic Acids Research, Webserver issue (43) doi: 10.1093/nar/gkv451]

PLOS authors have the option to publish the peer review history of their article (what does this mean?). If published, this will include your full peer review and any attached files.

Reviewer #1: No

Reviewer #2: No

Figure Files:

Data Requirements:

Reproducibility:

References:

---

## [Decision Letter · Decision Letter 1]

10 Oct 2023

Dear Dr. Sheiner,

Thank you very much for submitting your manuscript "Functional and biochemical characterisation of the Toxoplasma gondii succinate dehydrogenase complex" for consideration at PLOS Pathogens. As with all papers reviewed by the journal, your manuscript was reviewed by members of the editorial board and by several independent reviewers. The reviewers appreciated the attention to an important topic. Based on the reviews, we are likely to accept this manuscript for publication, providing that you modify the manuscript according to the review recommendations.

Both reviewers feel that previous concerns were adequately addressed, though there is a suggestion from reviewer #1 to make a minor alteration to the text regarding adding some discussion of data that is otherwise somewhat ignored. I agree that this would likely strengthen the manuscript.

With regard to citing unpublished data -- "Personal communication" citations have fallen out of favor for very good reason. That is what bioRxiv is for. If there is no preprint of unpublished data, it seems inappropriate to cite it. A reader has no way of assessing the relevant cited source for themselves, and at a later date has no way of knowing whether or not (or where) the data was eventually published. I would suggest removing this citation unless such a preprint now exists.

Sincerely,

Michael L Reese, PhD

Academic Editor

PLOS Pathogens

Margaret Phillips

Section Editor

PLOS Pathogens

Kasturi Haldar

Editor-in-Chief

PLOS Pathogens

orcid.org/0000-0001-5065-158X

Michael Malim

Editor-in-Chief

PLOS Pathogens

orcid.org/0000-0002-7699-2064

Both reviewers feel that previous concerns were adequately addressed, though there is a suggestion from reviewer #1 to make a minor alteration to the text regarding adding some discussion of data that is otherwise somewhat ignored. I agree that this would likely strengthen the manuscript.

With regard to citing unpublished data -- "Personal communication" citations have fallen out of favor for very good reason. That is what bioRxiv is for. If there is no preprint of unpublished data, it seems inappropriate to cite it. A reader has no way of assessing the relevant cited source for themselves, and at a later date has no way of knowing whether or not (or where) the data was eventually published. I would suggest removing this citation unless such a preprint now exists.

Reviewer Comments (if any, and for reference):

Reviewer's Responses to Questions

**Part I - Summary**

Reviewer #1: The authors have adequately addressed the prior critiques via responsive modifications. This manuscript will be a strong addition to the apicomplexan literature on ETC function. Two minor suggestions that might be addressed in the final manuscript are listed below.

Reviewer #2: In the reported work, Marina Silva et al verified that several novel subunits of the purported Toxoplasma mitochondrial complex II previously identified in complexomic studies are indeed components of a large mitochondrial complex and are required for succinate-ubiquinone oxidoreductase activity in the parasite, as expected for complex II components. The experiments addressing these points appear to be thorough and the results compelling.

**Part II – Major Issues: Key Experiments Required for Acceptance**

Reviewer #1: None

Reviewer #2: The authors have adequately addressed this reviewer's question regarding claimed potential of complex II to be a drug target in Toxoplasma and other apicomplexan parasites by appropriate modifications of the text and reference to pre-publication in vivo results from another well-regarded Group.

**Part III – Minor Issues: Editorial and Data Presentation Modifications**

Reviewer #1: 1. For the data in Figure 6B that indicate increased episomal SDH10-DY/AA protein levels upon Atc addition, can the authors extend the sentence in lines 354-355 to propose a possible biochemical basis for this increase? It seems odd to note this increase without any interpretation.

2. This reviewer appreciates the authors' dilemma in not being able to cite a formal van Dooren reference (line 371), since their manuscript is still in preparation. The editors will know best what format conforms with journal standards, but it looks like situations like this one require description as a "personal communication" and an accompanying letter from the relevant authors making the communication of unpublished results: https://journals.plos.org/plospathogens/s/submission-guidelines#:~:text=Do%20not%20cite%20the%20following,in%20a%20publicly%20available%20database.

Reviewer #2: In this revision the authors have addressed the points previously raised by this reviewer. However, in their citation of Giel Van Dooren and Soraya Zwahlen they should indicate in the text that it is a personal communication (i.e., not yet published).

S6 Fig caption does not list all the sequences for which alignments are shown, e.g., MPODD and SDH10. It also does not provide the names of the species included (shown as two-letter abbreviations in the figure). S7 Fig caption does not provide the name of the species designated as "Po" in one alignment.

PLOS authors have the option to publish the peer review history of their article (what does this mean?). If published, this will include your full peer review and any attached files.

Reviewer #1: No

Reviewer #2: **Yes: **Michael W Mather

Figure Files:

Data Requirements:

Reproducibility:

References:

---

## [Editor Report · Decision Letter 2]

26 Oct 2023

Dear Dr. Sheiner,

Thank you very much for submitting your manuscript "Functional and biochemical characterisation of the Toxoplasma gondii succinate dehydrogenase complex" for consideration at PLOS Pathogens. As with all papers reviewed by the journal, your manuscript was reviewed by members of the editorial board and by several independent reviewers. The reviewers appreciated the attention to an important topic. Based on the reviews, we are likely to accept this manuscript for publication, providing that you modify the manuscript according to the review recommendations.

The authors have adequately addressed all previous concerns. I apologize that I missed it with the last revision, but the MS data must be deposited into an appropriate repository and the accession numbers noted in manuscript, according to journal policies.

Sincerely,

Michael L Reese, PhD

Academic Editor

PLOS Pathogens

Margaret Phillips

Section Editor

PLOS Pathogens

Kasturi Haldar

Editor-in-Chief

PLOS Pathogens

orcid.org/0000-0001-5065-158X

Michael Malim

Editor-in-Chief

PLOS Pathogens

orcid.org/0000-0002-7699-2064

The authors have adequately addressed all previous concerns. I apologize that I missed it with the last revision, but the MS data must be deposited into an appropriate repository and the accession numbers noted in manuscript, according to journal policies.

Reviewer Comments (if any, and for reference):

Figure Files:

Data Requirements:

Reproducibility:

References:

---

## [Editor Report · Decision Letter 3]

27 Nov 2023

Dear Dr. Sheiner,

We are pleased to inform you that your manuscript 'Functional and biochemical characterisation of the Toxoplasma gondii succinate dehydrogenase complex' has been provisionally accepted for publication in PLOS Pathogens.

Best regards,

Michael L Reese, PhD

Academic Editor

PLOS Pathogens

Margaret Phillips

Section Editor

PLOS Pathogens

Kasturi Haldar

Editor-in-Chief

PLOS Pathogens

orcid.org/0000-0001-5065-158X

Michael Malim

Editor-in-Chief

PLOS Pathogens

orcid.org/0000-0002-7699-2064

Looks good to go. Though minor typo in your PRIDE accession sentence (first "in" should be "is").
---

## [Editor Report · Acceptance letter]

5 Dec 2023

Dear Dr. Sheiner,

We are delighted to inform you that your manuscript, "Functional and biochemical characterisation of the Toxoplasma gondii succinate dehydrogenase complex," has been formally accepted for publication in PLOS Pathogens.

Best regards,

Michael Malim

Editor-in-Chief

PLOS Pathogens

orcid.org/0000-0002-7699-2064